# Normalization of Language Embeddings for Cross-Lingual Alignment

**Prince Osei Aboagye[1], Yan Zheng[2], Chin-Chia Michael Yeh[2], Junpeng Wang[2],**
**Wei Zhang[2], Liang Wang[2], Hao Yang[2], Jeff M. Phillips[1]**
[1]University of Utah, [2]Visa Research
[1]{prince,jeffp}@cs.utah.edu
[2]{yazheng,miyeh,junpenwa,wzhan,liawang,haoyang}@visa.com

## Abstract

Learning a good transfer function to map the word vectors from two languages into a shared cross-lingual word vector space plays a crucial role in cross-lingual NLP. It is useful in translation tasks and important in allowing complex models built on a high-resource language like English to be directly applied on an aligned low resource language. While Procrustes and other techniques can align language models with some success, it has recently been identified that structural differences (for instance, due to differing word frequency) create different profiles for various monolingual embedding. When these profiles differ across languages, it correlates with how well languages can align and their performance on cross-lingual downstream tasks. In this work, we develop a very general language embedding normalization procedure, building and subsuming various previous approaches, which removes these structural profiles across languages without destroying their intrinsic meaning. We demonstrate that meaning is retained and alignment is improved on similarity, translation, and cross-language classification tasks.

## 1 Introduction

The standard multilingual NLP approaches typically do not jointly learn a single embedding, since words tend to cluster by language, and thus are not useful for translation and cross-lingual learning tasks. Rather, after learning individual embeddings, the usual approach is to map word vectors from multiple languages into a shared cross-lingual word vector space (Glavaš et al., 2019). This shared space creates a cross-lingual word embedding (CLWE) (Karan et al., 2020; Wang et al., 2019). These serve as a valuable tool for transferring data across different languages, understanding cross-linguistic differences, and cross-lingual transfer for downstream tasks, such as direct translation (Gouws et al., 2015; Heyman et al., 2017; Lample et al., 2018), cross-lingual information retrieval (Vulic & Moens, 2015), cross-lingual document classification (Klementiev et al., 2012), and cross-lingual dependency parsing (Guo et al., 2015; Ahmad et al., 2019).

A common element of almost all CLWE methods is the use of a rigid, orthogonal transformation mapping one embedding onto another so they inhabit a shared linguistic space. An orthogonal transformation is a special class of transformations that can be interpreted as the space of (in our case, high-dimensional) rotations around the origin, and allowing a mirror flip. This family of transformations preserves (a) linear and (b) angular properties. By linear properties, we mean that the straight-line Euclidean distance between elements is preserved, as are more powerful properties like analogies (e.g., Paris - France + Italy ≈ Rome). Angular properties refer to measuring angles between pairs of points (from the origin), and as a result, cosine distance is preserved. Given a correspondence between pairs of objects across two embeddings, the classic Procrustes method, provides a closed-form solution which minimizes the sum of Euclidean distances. If the vectors are all first normalized, then this also maximizes the sum of cosine similarities (Dev et al., 2021).

Under this framework, there has been a flurry of work significantly improving CLWE model performance along with two directions. Semi-supervised and unsupervised models make these approaches require less input, and more amenable to lower-resource languages. For example Bootstrap Procrustes (PROC-B) (Glavaš et al., 2019; Vulić et al., 2019) is semi-supervised in that it starts with a small pairwise correspondence (of 500-1000 words), aligns those to infer a larger correspondence,

and repeats applying Procrustes alignment. Methods like MUSE (Conneau et al., 2018a) are unsupervised, and use a GAN to estimate a correspondence before applying a Procrustes procedure.

The second direction is preprocessing the embeddings before applying the Procrustes alignment. These involve methods like removing the mean, removing principal components, and vector length normalization, discussed later. In principle, these methods aim to remove the geometry of data intrinsic to particular languages (but not shared across languages) while preserving similarity properties as assured by orthogonal alignment. The space of transformations allowed under orthogonal alignments is quite large, and we make the point that unless this data geometry is "normalized" it inhibits the alignment from optimizing over the entirety of this large space.

Finally, we note that methods like Canonical Correlation Analysis (CCA) (Faruqui & Dyer, 2014), Discriminative Latent Variable (DLV) (Ruder et al., 2018), and Ranking-based optimization (RC-SLS) (Joulin et al., 2018) have also been applied towards finding an orthogonal alignment (or pair of alignments) which minimizes a different optimization function – since the objective function may not align with sums of squared Euclidean or cosine distance (Conneau et al., 2018a; Smith et al., 2017). Unlike the others, the RCSLS method notably does not require a rigid transformation.

This paper is on embedding *preprocessing*, and is agnostic to the alignment used afterward.

**Our contribution.** This work proposes a new and general approach to preprocessing word embeddings, subsuming many previous approaches. The key is *Spectral Normalization* which regularizes the spectral properties of monolingual embeddings by setting all of the top singular vectors to have the same singular value. It leaves alone the smaller singular values; these capture important information and cannot be zeroed out, but making them the same value as the top singular vectors introduces too much noise. Spectral normalization already performs as well as the best previous approaches on alignment and translation tasks, and since it applies a fairly uniform stretching to the embeddings it does not distort monolingual similarity performance. Moreover, we show layering Spectral Normalization within an iterative sequence with also centering and vector length normalization improves results further. We demonstrate this improvement on the standard translation tasks, as well as on downstream cross-lingual use cases of document classification and natural language inference. We provide code at https://github.com/poaboagye/SpecNorm.

**Beyond CLWE.** This paper focuses on normalization for the alignment of word vector embeddings. This is for a couple of reasons. First, most relevant prior work does the same in the context of language translation, and so it allows for direct comparison. Second, these contexts have generally the largest embeddings, are trained on the most data, and are easy to interpret. However, as foundation models (Bommasani et al., 2021) become ubiquitous, many other settings induce similar challenges. For instance, alignment of various models from different data sources allows for improved ensemble embeddings (Dev et al., 2021), and this could extend to image (Frome et al., 2013; Kiela & Bottou, 2014) or graph embeddings (Grover & Leskovec, 2016; Perozzi et al., 2014). Or in aligning embeddings of spatial data (Jenkins et al., 2019) or of merchants (Wang et al., 2021), one may want to align data from different geographic regions for tasks in transfer learning and domain adaptation. As an example of generality, we demonstrate an improvement via our normalization approaches in aligning genomic data embeddings (Demetci et al., 2020) in Appendix G.

## 2    EXISTING METHODS FOR ORTHOGONAL VECTOR SPACES ALIGNMENT

In an embedded representation of a set of $n$ words, each $i$th word is associated with a vector $a^i \in \mathbb{R}^d$, so $A = \{a^1, \ldots, a^n\}$ is the set of $n$ word vector representation. These vector representations (derived by methods like word2vec (Mikolov et al., 2013a), GloVe (Pennington et al., 2014), or FastText (Bojanowski et al., 2017)) are chosen so words with similar pairwise cosine similarity are found in the similar local context in large text corpora on which they are trained. Higher-level linear structure is shown to emerge, such as concept subspaces and analogies (Mikolov et al., 2013).

The focus of this paper is on aligning embeddings of two languages $L_1$ and $L_2$. Each embedding $A_{L_1}$ and $A_{L_2}$, is only designed to ensure pairwise relationships between its word vectors, but the actual coordinates of those vectors do not have any explicit meaning. Yet, previous work has clearly demonstrated that there exists significant overall structural similarity, and alignment seeks to make correspondences between those structures for translation and joint understanding.

## 2.1 PRE-PROCESSING EMBEDDINGS BEFORE ORTHOGONAL ALIGNMENT

It turns out directly aligning embeddings from two languages (even using the "optimal" Procrustes solution) does not provide the best possible joint embedding for translation tasks. While word meaning appears to hold a similar structure, languages have other properties such as differing word frequency, and this for instance leads to more frequent words having longer vectors in embeddings. This extra language-specific structure tends to interfere with alignment. As a result, a number of techniques have been developed to "normalize" the embeddings before Procrustes (or other) alignment. This, in some sense, allows the word meaning to dominate the optimization tasks without other confounding factors. We review the most common normalization approaches.

**Mean Centering (C)** subtracts the mean of all vectors in an embedding from each vector in that embedding. The result is that the mean of all vectors is $0$. This is a rigid transformation, and so does not change the Euclidean distance between any pair of points in an embedding, and also preserves any linear property like analogies (e.g., Paris - France + Italy $\approx$ Rome). Dev et al. (2021) points out that this is the first step (followed by the Procrustes orthogonal transformation) to minimize the sum of squared Euclidean distances among paired words, under any rigid transformation. However, this *does change* the cosine distance between pairs of points.

**Length Normalization (L)** makes each vector have a 2-norm equal to $1$, but retains its direction from the origin (Artetxe et al., 2016; Xing et al., 2015). This preprocessing step does not change the cosine distance between any pair of points in an embedding. But, it *does change* the Euclidean distance between pairs of points.

Despite these contrasting goals, these two normalizations each turn out to be individually effective in regularizing the geometry of the embeddings, and allow for better CLWE. Zhang et al. (2019), realized doing both was even more effective, and showed that iterating these two steps achieves the state-of-the-art way to preprocess, we denote as **I-C+L**. Iterative Normalization eventually transforms monolingual word embeddings to have unit-length and zero-mean simultaneously (in practice they terminate this iterative process after a few steps before it achieves these two goals exactly).

**PCA Removal (PR)** computes the principal component analysis (PCA) of an embedding, and then projects away from the direction of the top principal component, removing it. Mu & Viswanath (2018) observed these directions typically do not encode essential semantic relationships between words but rather align strongly with word frequency; and PCA removal before alignment led to improved performance on several tasks. Sachidananda et al. (2021) also showed this preprocessing improved the BLI task.

## 2.2 SPECTRAL STATISTICS OF EMBEDDINGS

Dubossarsky et al. (2020) recently documented how cross-lingual alignment is strongly affected by the spectral statistics of monolingual embeddings. We stack the embedded vectors $a^i \in \mathbb{R}^d$ as rows in a $n \times d$ matrix $A \in \mathbb{R}^{n \times d}$. The SVD decomposes $A$ into $U \Sigma V^\top$ where $U$ and $V$ contain the left and right singular vectors, and the singular values $\sigma_1 \geq \sigma_2 \geq \ldots \geq \sigma_d \geq 0$ are on the diagonal of $\Sigma$. The *effective rank* of $A$ is a smoother analog to rank (when there is noise in low rank components), defined $\mathsf{er}(A) = e^{H(\Sigma)}$ where $H(\Sigma) = -\sum_{i=1}^{d} \bar{\sigma}_i \log \bar{\sigma}_i$ with $\bar{\sigma}_i = \sigma_i / \sum_{i=1}^{d} \sigma_i$. The *effective condition number* $\kappa_{\mathsf{eff}}(A) = \sigma_1 / \sigma_{\mathsf{er}(A)}$, which replaces the numerator (of condition number, $\sigma_d$) with the more robust singular value at the effective rank. This is desired to be small in stable data sets. The *joint effective condition number* measures the harmonic mean of the effective condition number across two matrices $A, A'$ as $\mathsf{ECOND\text{-}HM}(A, A') = \frac{2\kappa_{\mathsf{eff}}(A)\kappa_{\mathsf{eff}}(A')}{\kappa_{\mathsf{eff}}(A)+\kappa_{\mathsf{eff}}(A')}$. The *singular value gap* measures how similar the singular value sequences are between two matrices as $\mathsf{SVG}(A, A') = \sum_{i=1}^{d} (\log \sigma_i - \log \sigma_i')^2$. These should both be smaller, for more comparable data sets.

Dubossarsky et al. (2020) applied these to monolingual embeddings and demonstrated that the performances of several CLWE methods were closely tied to these spectral properties. Basically embeddings align better if they are better jointly conditioned, especially measured via joint effective condition number and the singular value gap. Contextual embeddings have also been shown to suffer from similar challenges (Ethayarajh, 2019; Xu & Koehn, 2021). Motivated by this idea, we propose methods that spectrally normalize embeddings improving these statistics while retaining intra-embedding meaning.

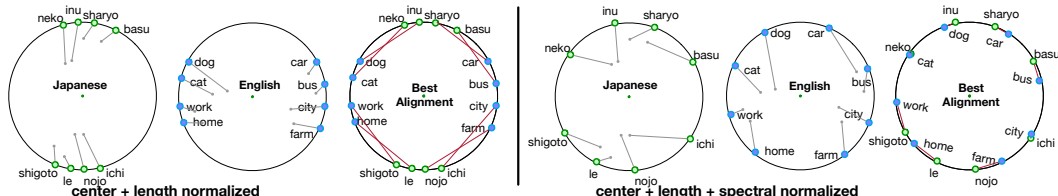

Figure 1: Illustration on toy $2d$ data showing potential advantage of Spectral Normalization beyond centering and length normalization. After center and length normalization all points are on the unit circle, centered at the origin, but may have uneven clustering. The addition of Spectral Normalization can disperse these clusters allowing improved alignment.

## 3 NEW METHOD: SPECTRAL NORMALIZATION

Unit-length and zero-mean normalization makes embedding vectors from a language lie on a hypersphere with the center of the hypersphere centered at the origin. However, this does not take into account how the word embeddings vectors are clustered on the hypersphere. See example in Figure 1 where despite centering and length normalization, words are grouped differently across languages, and this prevents a close alignment. Approaches like PCA removal and mean centering have the effect of reducing the top principal component or top singular vector. As a result, if the spectral properties are extreme, it can help regularize them. However, this approach can be blunt. PCA removal makes the top singular value exactly 0, so the condition number becomes infinite.

To this effect, we propose a new algorithm Spectral Normalization that more gently regularizes the spectral properties of word embeddings; see Algorithm 1. We will then combine it with other approaches to again ensure the embedding vectors lie on the unit sphere. In Figure 1, this approach spreads out words on the unit sphere without reordering them, and allows for a closer alignment.

---

**Algorithm 1** Spectral Normalization ($\mathsf{SpecNorm}(A, \beta)$)

1: Compute $\mathsf{svd}(A) = U\Sigma V^\top$; Let $D \in \mathbb{R}^{d \times d}$ be a diagonal matrix.
2: Compute $\eta = \sqrt{\|A\|_F^2/d}$, where $d$ is the dimension of the word embedding
3: **for** $i = 1, \ldots, d$ **do**
4:     **if** $(\Sigma_{ii} > \beta\eta)$ **then** $D_{ii} \leftarrow \Sigma_{ii}/(\beta\eta)$
5:     **else** $D_{ii} = 1$.
6: **return** $AVD^{-1}$

---

This updates part of the spectral properties of embedding $A$ as a whole, using a parameter $\beta \geq 0$ (fixed later as $\beta = 2$ via cross-validation). Based on an average of singular values $\eta = \sqrt{\|A\|_F^2/d}$, if a value is above $\beta$ times that average, it adjusts it to $\beta\eta$. Hence, all of the top directions are given the same singular value. Otherwise, if it is below $\beta\eta$, it is considered a minor effect (some are quite small, and fairly noisy), and it is left alone. If these small ones are completely zeroed out, the critical information within is destroyed. However, if these small ones are given the same value (i.e., $\beta\eta$) then components which do not contribute to the most prevalent aspects of a vectors similarity are given more importance, and we observe that the usefulness of the embedding decreases.

**Iterative Spectral Normalization.** Spectral normalization makes the most sense (see Appendix J) in a setting where the vectors are already centered, and also unit length. While SpecNorm does not change the center of the data, it does not maintain the length of individual vectors. As such, we advocate combining these methods into a single iterative algorithm: I-C+SN+L as in Algorithm 2.

---

**Algorithm 2** Iterative Spectral Normalization with C+L normalization (I-C+SN+L($A$, #Iter))

1: **for** #Iter steps **do**
2:     $A \leftarrow$ Center $A$
3:     $A \leftarrow$ SpecNorm($A$)
4:     $A \leftarrow$ Unit length normalization of $A$
5: **return** $A$

---

We observe in Figure 2 that this process significantly improves the spectral properties compared to any other approach. Without preprocessing (None), the languages (EN: English, DE: German, HI: Hindi, JA: Japanese shown) have large effective condition numbers – indicating that there is a large disparity between meaningful singular values. Note the y-axis is in the log scale. Hence, aligning these languages without normalization would likely restrict alignment among top singular vectors, not allowing enough degree of freedom to align corresponding words.

In contrast, after preprocessing when these values are more uniform, rotations among the dimensions containing the top principal components will not have an influence on the data distribution, and can fully optimize the alignment between words. Moreover, Figure 2 shows that I-C+SN+L most decreases the effective condition number, joint effective condition number, and singular value gap. Further, these values are fairly uniform across languages, despite great variation beforehand (as shown with None). In fact, I-C+SN+L is much more effective than other methods.

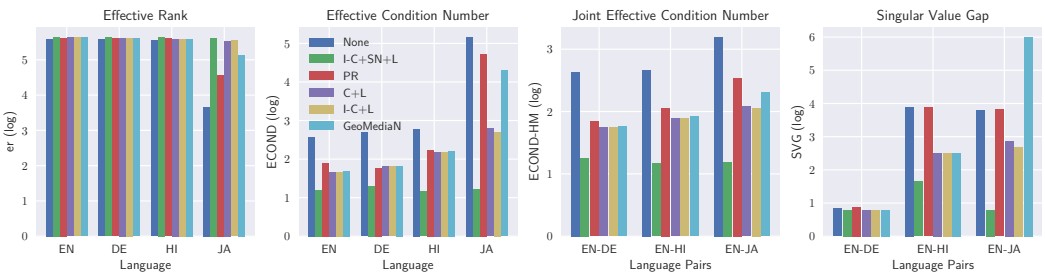

Figure 2: Spectral Measures of four (4) monolingual word embeddings, before (None) and after applying various normalization methods.

**Is Iteration Necessary?**   While alternating optimization is a common paradigm, is it necessary or useful to achieve these normalization goals? There may be many solutions which achieve length, center, and spectral normalization simultaneously. In fact, we observe that "centering" with the geometric median (Chatelon et al., 1978; Eyster et al., 1973; Overton, 1983) and then length normalizing vectors (we call this GeoMediaN, and detail it in Appendix A) achieves length and center normalization without iteration. However, as we observe, it under-performs I-C+L.

## 4    EXPERIMENTAL ANALYSIS

We provide an evaluation of our proposed preprocessing methods using eight (8) language embeddings pre-trained on Wikipedia (Bojanowski et al., 2017) of each language: Croatian (HR), English (EN), Finnish (FI), French (FR), German (DE), Italian (IT), Russian (RU), and Turkish (TR). We use the 300-dimensional fastText (Bojanowski et al., 2017) embeddings, and all vocabularies are trimmed to the 200K most frequent words.

**Alignment evaluation tasks: BLI**   We evaluate and compare our proposed preprocessing methods mostly on the Bilingual Lexicon Induction (BLI) task, a word translation task. BLI has become the de facto evaluation task for CLWE models. For words in the source language, this task retrieves the nearest neighbors in the target language after alignment to check if it contains the translation. It reports the mean average precision (MAP) (Glavaš et al., 2019), which is equivalent to the mean reciprocal rank (MRR), of the translation. Unless stated otherwise, reported values on baseline methods are taken from (Glavaš et al., 2019), and use the Google Translate (GTrans) dictionary from (Glavaš et al., 2019). We trained (aligned) using 1k, 3k and 5k source words and evaluated (tested) on separate 2k source test queries, unless noted otherwise.

**Alignment Algorithms.**   We evaluated and compared the result of several supervised rigid-transformation CLWE models on the evaluation benchmarks using our proposed methods. All have publicly available codes, links are found in the reference citation. These include Canonical Correlation Analysis (CCA) (Faruqui & Dyer, 2014), Procrustes (PROC) (Artetxe et al., 2016; Smith et al., 2017; Xing et al., 2015; Dev et al., 2021), Bootstrapping Procrustes (PROC-B) (Glavaš et al., 2019), and Discriminative Latent-Variable (DLV) (Ruder et al., 2018), as discussed in Section 1.

We also consider Ranking-Based Optimization (RCSLS) (Joulin et al., 2018) which is not a rigid alignment. In a few places, we also compare with VECMAP (Artetxe et al., 2018) as an example of an unsupervised alignment process. This should only use the geometry of the global embedding structure, e.g., derived from the natural ontology, and our normalization method still helps.

## 4.1 Hyperparameter Tuning

Our main proposed algorithm I-C+SN+L has a few simple parameters. To avoid overfitting, we choose these through cross-validation on English (EN) and a held-out set of five (5) languages Hindi (HI), Russian (RU), Chinese (ZH), Japanese (JA), Turkish (TR). Ten (10) Language pairs of the form EN-X and X-EN were considered. The hyperparameters $\beta \in \{1, 2, 3, 4, 5\}$ and #Iter (number of iterations) $\in \{1, 2, 3, 4, 5\}$ were fine-tuned for I-C+SN+L.

We used the publicly available MUSE translation dictionary (Conneau et al., 2018a) for hyperparameter tuning. The Procrustes alignment algorithm was trained on 5k source words and evaluated on 1.5k source test queries. We reported the mean average precision (MAP) in Table 1 for $\beta \in \{1, 2, 3, 4, 5\}$ and with #Iter $\in \{1, 2, 3, 4, 5\}$. We observe the value of $\beta = 2$ was consistently the best threshold (although any $\beta \geq 2$ performed similarly). However, the result did not change much with respect to the number of iterations.

Table 1: Cross-Validation for Hyperparameter Tuning: MAP after Procrustes for 10 language pairs.

| $\beta$ | #Iter=1 | #Iter=2 | #Iter=3 | #Iter=4 | #Iter=5 |
|---|---|---|---|---|---|
| 1 | 0.363 | 0.340 | 0.328 | 0.322 | 0.317 |
| 2 | 0.385 | **0.386** | **0.386** | **0.386** | **0.386** |
| 3 | 0.381 | 0.384 | 0.384 | 0.384 | 0.384 |
| 4 | 0.381 | 0.382 | 0.382 | 0.382 | 0.382 |
| 5 | 0.380 | 0.381 | 0.381 | 0.381 | 0.381 |

The tie between the #Iter hyperparameter was broken using their performance on 13 English word similarity benchmarks; details in Appendix D. In Table 2 reports the average Spearman rank coefficient score on the tasks (None means no normalization). $(\beta, \text{#Iter}) = (2, 5)$ achieved the highest score. So hereafter, we applying I-C+SN+L with the hyperparameter $(\beta, \text{#Iter}) = (2, 5)$.

Table 2: Monolingual word similarity task; Average Spearman rank coefficient

| None | $(\beta, \text{#Iter}) = (2, 2)$ | $(\beta, \text{#Iter}) = (2, 3)$ | $(\beta, \text{#Iter}) = (2, 4)$ | $(\beta, \text{#Iter}) = (2, 5)$ |
|---|---|---|---|---|
| 0.651 | 0.67077 | 0.67101 | 0.67108 | **0.67111** |

Note that the proposed approach (I-C+SN+L) only increased this score for these similarity tasks, so showed no signs of distorting inherent information. Although Spectral Normalization does not exactly preserve the linear properties or angular properties (as centering and length normalization do, one each, respectively), it does not suffer ill effects. We hypothesize this is because it is somewhat uniformly stretching words along with the major modes of variation, and is effectively removing information not relevant to meaning, like frequency. This benign effect is in contrast to other spectral adjustments (removing small singular values, or setting all to the same value); see Appendix D.

## 4.2 BLI Performance across Normalization and Alignment Algorithms

We compare and evaluate the BLI performance (MAP) of various normalization algorithms from previous works to our proposed algorithms. Using the MUSE translation dictionary, we trained CCA, PROC, PROC-B and RCSLS on 5k source words and evaluated on 1.5k source test queries. The following normalization algorithms were used in the comparison analysis: PR (PCA Removal) (Mu & Viswanath, 2018), GeoMediaN (Geometric Median Normalization), C+L (Mean centering and Length normalization, 1 round), I-C+L (Iterative Mean centering and Length normalization, 5 rounds) (Zhang et al., 2019), SN (Spectral Normalization, 1 round), C+SN+L (Mean centering, Spectral Normalization and Length normalization, 1 round), and I-C+SN+L (Iterative Mean

centering, Spectral Normalization and Length normalization, 5 rounds). Specifically, we evaluated 18 language pairs, i.e., English (EN) from/to Bulgarian (BG), Catalan (CA), Czech (CS), German (DE), Spanish (ES), Korean (KO), Thai (TH) and Chinese (ZH), separate from hyperparameter tuning. The average is reported in Table 3, all results are in Appendix J. For almost all algorithms I-C+SN+L achieves the best scores (and especially on $\mathbf{X}_L - \mathbf{EN}$, often considerably better). The only exceptions are on non-rigid RCSLS when C+SN+L (with no iteration) or just SN (without C+L) performs slightly better. So, Spectral Normalization, and in particular I-C+SN+L, is shown as the best way to normalize languages before alignment.

Table 3: BLI performance (MAP) on aligning $\mathbf{EN} - \mathbf{X}_L$ and $\mathbf{X}_L - \mathbf{EN}$

| Normalization | Methods : $\mathbf{EN} - \mathbf{X}_L$ | | | | Methods : $\mathbf{X}_L - \mathbf{EN}$ | | | |
| | CCA | PROC | PROC-B | RCSLS | CCA | PROC | PROC-B | RCSLS |
|---|---|---|---|---|---|---|---|---|
| None | 0.358 | 0.365 | 0.377 | 0.394 | 0.398 | 0.399 | 0.405 | 0.428 |
| PR | 0.394 | 0.391 | 0.404 | 0.373 | 0.434 | 0.430 | 0.442 | 0.425 |
| GeoMediaN | 0.393 | 0.391 | 0.400 | 0.379 | 0.433 | 0.432 | 0.440 | 0.429 |
| C+L | 0.393 | 0.394 | 0.408 | 0.404 | 0.439 | 0.437 | 0.445 | 0.464 |
| I-C+L | 0.394 | 0.395 | 0.410 | 0.406 | 0.439 | 0.438 | 0.448 | 0.460 |
| SN | 0.391 | 0.394 | 0.408 | 0.405 | 0.440 | 0.438 | 0.451 | **0.468** |
| C+SN+L | 0.395 | 0.396 | 0.413 | **0.407** | 0.444 | 0.444 | 0.458 | 0.466 |
| I-C+SN+L | **0.396** | **0.398** | **0.414** | 0.406 | **0.445** | **0.446** | **0.461** | 0.466 |

We also compute the average BLI MAP score across all 28 language pairs for more direct comparison to prior work (Glavaš et al., 2019), summarized in Table 4 and Appendix E. All results are in Appendix L. We compare I-C+SN+L (denoted with SN) against no normalization on various dictionary sizes: 1k, 3k and 5k source words and evaluated on 2k source test queries. *In all cases, I-C+SN+L consistently improves over the baseline; see Appendix E.* This includes improvement over RCSLS which is non-rigid, so in principle could "learn" adjustments similar to our normalization in the process of alignment. We also tested on VECMAP, an unsupervised approach; I-C+SN+L preprocessing also improves this result from 0.375 to 0.410.

Table 4: Summary of BLI performance (MAP), average scores for all 28 language pairs. No normalization results from (Glavaš et al., 2019), against I-C+SN+L (denoted SN).

| Dict | CCA | $\text{CCA}^{\text{SN}}$ | PROC | $\text{PROC}^{\text{SN}}$ | PROC-B | $\text{PROC-B}^{\text{SN}}$ | DLV | $\text{DLV}^{\text{SN}}$ | RCSLS | $\text{RCSLS}^{\text{SN}}$ |
|---|---|---|---|---|---|---|---|---|---|---|
| 1k | .289 | **.314** | .299 | **.326** | .379 | **.407** | .289 | **.332** | .331 | .331 |
| 3k | .378 | **.401** | .384 | **.408** | .398 | **.415** | .381 | **.429** | .415 | **.427** |
| 5k | .400 | **.423** | .405 | **.429** | – | – | .403 | **.452** | .437 | **.460** |

Table 5: BLI performance (MAP) on aligning Cross-lingual **Contextual** Embedding, $\mathbf{EN} - \mathbf{X}_L$

| Embedding | Normalization | EN-AR | EN-DE | EN-NL | Avg |
|---|---|---|---|---|---|
| FastText | None | 0.256 | 0.357 | 0.477 | 0.363 |
| Type-level | None | 0.501 | 0.441 | 0.540 | 0.494 |
| FastText | I-C+L | 0.284 | 0.372 | 0.493 | 0.383 |
| Type-level | I-C+L | 0.510 | 0.449 | 0.543 | 0.501 |
| FastText | I-C+SN+L | 0.280 | 0.375 | 0.499 | 0.385 |
| Type-level | I-C+SN+L | **0.525** | **0.450** | **0.544** | **0.506** |

**Normalizing contextual type-level embeddings.** Table 5 compares the impact of Iterative Normalization and Spectral Normalization on the BLI performance on aligning Cross-lingual Contextual Embedding. We follow the implementation details from Xu & Koehn (2021) for learning representatives within BERT (see Appendix C for more details). Then the learned normalization can be viewed as a composition to the functional embedding, so is compatible with downstream uses. Our proposed normalization algorithm, Iterative Spectral Normalization (I-C+SN+L), clearly outperforms Iterative Normalization (I-C+L) on the BLI task for aligning Contextual Embeddings.

This buttresses the claim made by Xu & Koehn (2021) that improving the degree of isomorphism between contextual embeddings spaces enhances the quality of learned alignment and subsequently a better performance on the BLI task.

## 4.3 DOWNSTREAM TASKS

We conclude by demonstrating that Spectral Normalization not only improves in direct translation tasks, but also captures an important global structure that generalizes from a high resource language (i.e., English, EN) to lower resource languages. In both examples, a powerful classifier is trained on the EN embedding (after normalization), and then we demonstrate that after a lower resource language (e.g., German, DE) has been normalized and aligned the analysis task can be directly applied to that language. In particular, adding our normalization (I-C+SN+L) dramatically improves the results over not doing that step, and typically improves on I-C+L normalization.

**Cross-lingual Document Classification (CLDC).** The CLDC task builds a topic classification using a language model on a high resource language (in our case English EN) across 15 topics. The TED CLDC corpus assembled by Hermann & Blunsom (2014) was used for training and evaluation. Following Glavaš et al. (2019), a simple CNN was used to train. Table 6 summarizes the average F1-score for all topic classifiers on 5 language pairs. The CLWEs induced by PROC[SN], PROC-B[SN], DLV[SN], and RCSLS[SN] (using I-C+SN+L) outperformed the baseline result (with no normalization) on the CLDC task, greatly improving the best average score from $0.421$ to $0.461$, and improving over I-C+SN+L for all alignments except RCSLS. Glavaš et al. (2019) used only 12 of 15 topics, but could not confirm which, so we re-ran all baselines using all 15 topics.

Table 6: CLDC performance (micro-averaged $F_1$ scores). Cross-lingual transfer EN–X

| Model | Dict | EN-DE | EN-FR | EN-IT | EN-RU | EN-TR | Avg |
|---|---|---|---|---|---|---|---|
| PROC | 5k | .366 | .258 | .338 | .288 | .278 | .306 |
| PROC[I-C+L] | 5k | .452 | .325 | .427 | .521 | .479 | .440 |
| PROC[I-C+SN+L] | 5k | .436 | .366 | .427 | .517 | .511 | **.451** |
| PROC-B | 3k | .364 | .304 | .299 | .336 | .317 | .324 |
| PROC-B[I-C+L] | 3k | .478 | .341 | .403 | .527 | .506 | .451 |
| PROC-B[I-C+SN+L] | 3k | .448 | .396 | .423 | .522 | .517 | **.461** |
| DLV | 5k | .419 | .336 | .397 | .493 | .458 | .421 |
| DLV[I-C+L] | 5k | .434 | .313 | .377 | .464 | .489 | .415 |
| DLV[I-C+SN+L] | 5k | .433 | .323 | .406 | .499 | .472 | **.427** |
| RCSLS | 5k | .466 | .397 | .403 | .403 | .406 | .415 |
| RCSLS[I-C+L] | 5k | .445 | .514 | .529 | .443 | .443 | **.474** |
| RCSLS[I-C+SN+L] | 5k | .468 | .500 | .443 | .488 | .394 | .459 |

**Cross-lingual Natural Language Inference (XNLI).** We evaluated the CLWE on a cross-lingual natural language inference (XNLI) task. We used a multi-lingual XNLI corpus created by Conneau et al. (2018b), which is a collection of sentence pairs from the English MultiNLI corpus (Williams et al., 2018) translated into 14 languages. The MultiNLI corpus contains 433k sentence pairs with the labels entailment, contradiction, and neutral. The intersection between XNLI languages and BLI languages results in four XNLI evaluation pairs: EN-DE, EN-FR, EN-TR, and EN-RU. We use the training setup in Glavaš et al. (2019) with the Enhanced Sequential Inference Model (Chen et al., 2017) on English after normalization. First, we aligned normalized versions of each language onto

the normalized EN embedding to obtain the shared cross-lingual embedding. Then we used the 5k test pairs from the XNLI corpus to evaluate each language alignment. Table 7 shows the result for PROC, PROC-B, and RCSLS alignments (DLV and VECMAP transform the EN embedding in the process, so were omitted). We compare I-C+L and our I-C+SN+L normalization against the same procedure *without* normalization from Glavaš et al. (2019). As in other experiments, our normalization improves the average test accuracy with each alignment approach.

Table 7: XNLI performance (test set accuracy)

| Model | Dict | EN-DE | EN-FR | EN-TR | EN-RU | Avg |
|---|---|---|---|---|---|---|
| PROC | 5k | .607 | .534 | .568 | .585 | .574 |
| PROC$^{\text{I-C+L}}$ | 5k | .589 | .608 | .536 | .581 | .579 |
| PROC$^{\text{I-C+SN+L}}$ | 5k | .611 | .638 | .542 | .596 | **.597** |
| PROC-B | 3k | .615 | .532 | .573 | .599 | .580 |
| PROC-B$^{\text{I-C+L}}$ | 5k | .602 | .636 | .537 | .595 | .593 |
| PROC-B$^{\text{I-C+SN+L}}$ | 3k | .624 | .638 | .548 | .601 | **.603** |
| RCSLS | 5k | .390 | .363 | .387 | .399 | .385 |
| RCSLS$^{\text{I-C+L}}$ | 5k | .514 | .490 | .490 | .526 | .505 |
| RCSLS$^{\text{I-C+SN+L}}$ | 5k | .499 | .482 | .504 | .556 | **.510** |

## 5 CONCLUSION & DISCUSSION

We introduce a new way to normalize embeddings, based on spectral normalization, for use in creating cross-lingual word embeddings. Our approach generalizes previous approaches, and when used to individually preprocess monolingual embeddings, it allows alignment procedures to find better alignments: resulting in improved performance on direct translation tasks as well as cross-lingual topic classification and natural language inference tasks. Moreover, we demonstrate this improvement is very broadly useful; it holds in contextual embeddings as well as on embeddings of non-language data (on genomic data in Appendix G).

**Limitations.** Our proposed preprocessing method heavily relies on the *approximately isomorphic* assumption of the two monolingual embedding spaces to be aligned (Mikolov et al., 2013b; Ruder et al., 2019). Under this condition, it is easier to learn a robust linear map between two monolingual embedding spaces that are *approximately isomorphic* than spaces that are not. Recent, lines of work (Søgaard et al., 2018; Nakashole & Flauger, 2018; Patra et al., 2018) have questioned this *approximately isomorphic* assumption and have shown that it does not hold in general. Also, it hinders the performance of some CLWE methods. The goal of our proposed preprocessing method is to canonically preprocess the two monolingual embedding spaces to improve their isomorphism, and we show it is generally effective.

CLWE methods such as Joint CLWE methods (Wang et al., 2019) and Cross-lingual Anchoring (Ormazabal et al., 2021) departs from the *approximately isomorphic* assumption. They learn new embeddings from scratch, using a different objective function that encourages the alignment of representations or the vector embeddings of the words. As such our proposed preprocessing method does not apply. These methods, however, are less flexible and portable than alignment-based approaches, since they need to learn a new embedding for each pair of languages, and would need to re-train if only one is updated or a different or multiple pairs are considered. We have a preliminary comparison with Ormazabal et al. (2021) in Appendix F and show our methods are competitive.

## 6 ETHICS STATEMENT

The vast majority of NLP research and cutting-edge advancements are in English. This disadvantages those who primarily operate in other languages, with less developed models, or less data to train models. As large language models are the cornerstone of most NLP research and development in English, one of this work's main goals is to inexpensively port these advances to other languages, and those who use them. This will help unlock this technology to many around the world. As with most models, this accuracy and improvement may vary across tasks and languages.

While language models have many positive use cases including improving accessibility, better recommendations, and increased automation, they have some negative effects as well. These include requiring potentially large computational and hence environmental costs, encoding and exacerbating bias, and aiding in automatically generating fake or deceitful content. While this paper is unlikely to change the *desire* to use embeddings, it aims to reduce the burden of use and increase the effectiveness in lower-resource settings. And in particular to port models trained in English to other languages. This would reduce the cost of retraining in other languages if the English model can be reused, easing environmental costs. We support the maturing efforts in attenuating bias in all such embeddings. And while we acknowledge the possibility of this work aiding in the creation of deceitful content and the harm it can cause, we believe the many benefits outweigh the harms.

## 7 REPRODUCIBILITY STATEMENT

All existing methods are compared with publicly available codes with publicly available data, with links above or in references. The exception is code for CLDC and XNLI is shared by Glavaš et al. (2019). Everything was run with default parameters; the exception is RCSLS, where we follow the suggested hyperparameter selection strategy (Joulin et al., 2018) (with learning rate in $\{1, 10, 25, 50\}$ and epoch number in $\{10, 20\}$). Our new code for SpecNorm is in Appendix H and here https://github.com/poaboagye/SpecNorm.

ACKNOWLEDGMENTS

We thank our support from NSF IIS-1816149, CCF-2115677, and Visa Research.

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

APPENDIX

## A  GEOMETRIC MEDIAN NORMALIZATION

Iterative Normalization converges towards individual word embeddings having a unit length and each monolingual embedding having a zero mean. Zhang et al. (2019) showed that iterating solutions for these distinct goals will eventually converge to a solution which satisfies both.

In this paper, we observe that both goals can be done in one shot without iterating – by solving the Fermat-Weber problem (Maxwell, 1966; Love et al., 1988). This dates to the 17th century, and corresponds with identifying the geometric median of a point set. Formally, the goal is a point $x^* \in \mathbb{R}^d$ that minimizes the sum of distances from $n$ anchor points $\{a^1, \ldots, a^n\} \subset \mathbb{R}^d$ which are not colinear:

$$x^* = \min_x \sum_{i=1}^n \|a^i - x|. \tag{1}$$

Several methods (Chatelon et al., 1978; Eyster et al., 1973; Overton, 1983) have been proposed; the most popular is the Weiszfeld's algorithm (Weiszfeld, see below for details). It is folklore that the solution $x^*$ satisfies that $0 = \sum_{i=1}^n \frac{a^i - x^*}{\|a^i - x^*\|}$; we do not know of a written proof, so we prove this for completeness.

**Theorem A.1.** *If $x \in \mathbb{R}^d$ is distinct from all the given anchor points, $a^i$, then $x$ is the geometric median is true if and only if ($\iff$)*

$$0 = \sum_{i=1}^n \frac{a^i - x}{\|a^i - x\|} \tag{2}$$

*Proof.* ($\Rightarrow$) Suppose $x \in \mathbb{R}^d$ is distinct from all the given anchor points, $a^i$ and $x$ is the geometric median such that $x = \widetilde{T}(x)$ (see A.1) then

$$\nabla f(x) = \sum_{i=1}^n \frac{a^i - x}{\|a_i - x\|} = 0.$$

($\Leftarrow$) Suppose $x \in \mathbb{R}^d$ is distinct from all the given anchor points, $a^i$ and $x$ is the unique optimal solution of equation 1 such that $\nabla f(x) = 0$ then solving for $x$ while ignoring the dependency $x$ in $\|a^i - x\|$ yields:

$$x = \frac{\sum_{i=1}^n \|a^i - x\|^{-1} a^i}{\sum_{i=1}^n \|a^i - x\|^{-1}}$$

which is the geometric median. $\qquad\square$

Using this characteristic of the geometric median, we can simultaneously enforce monolingual word embeddings to have unit-length and zero-mean in just one step. This can be done using the Geometric Median normalization (GeoMediaN) algorithm (as Algorithm 3). Given a monolingual word embedding $A$, we compute the geometric median $x^*$, and "center" the data on this point, and unit length normalizes the centered embedding.

---

**Algorithm 3** Geometric Median Normalization: GeoMediaN($A$)

1: $x^* \leftarrow$ Weiszfeld($A$)
2: **for all** $a^i \in A$ **do** $a^i \leftarrow \frac{a^i - x^*}{\|a^i - x^*\|}$
3: **return** $A$

---

After these steps, all vectors are unit length, and because of the folklore property (Theorem A.1), the mean of those points is also $0$. As a result, we can state the following property.

**Theorem A.2.** *The output of* GeoMediaN(A) *is centered and length normalized.*

Despite the Geometric Median Normalization algorithm's ability to enforce unit-length and zero-mean in just one step, we observe that it does not perform especially well on the BLI task. Both GeoMediaN and I-C+L achieve one of many solutions which achieve these joint goals.

### A.1 WEISZFELD ALGORITHM

The Weiszfeld algorithm is an iterative method for finding the geometric median of a set of points in Euclidean space based on the reformulation of a stationary point that satisfies $\nabla f(x) = 0$.

If iteration function $T : \mathbb{R}^d \to \mathbb{R}^d$ is defined by:

$$
T(x) = \begin{cases} \widetilde{T}(x) = \dfrac{\sum_{i=1}^{n} \left\| a^i - x \right\|^{-1} a^i}{\sum_{i=1}^{n} \left\| a^i - x \right\|^{-1}} & if \quad x \quad \notin \quad \left\{ a^1, \ldots a^n \right\} \\ \\ a^i & if \quad x \quad = \quad a^i \ , \ i = 1, \ldots, n \end{cases}
\tag{3}
$$

then the Weiszfeld algorithm is:

$$
x^{k+1} = T\left(x^k\right), \ k \in \mathbb{N}
\tag{4}
$$

where $x^0 \in \mathbb{R}^d$ is a starting point. When the current iterate, $x^k \notin \left\{ a^1, \ldots a^n \right\}$, $T\left(x^k\right) = \widetilde{T}\left(x^k\right)$; else, if $x^k = a^i$, then $T\left(x^k\right) = a^i$.

The Weiszfeld algorithm is presented in Algorithm 4 below:

---
**Algorithm 4** Weiszfeld algorithm (WA)

---
**Input**: Anchor points, $\left(a^1, \ldots a^n\right)$, $x^0 \in \mathbb{R}^d$ and $\epsilon > 0$
1: $k \leftarrow 0$
2: **while** True **do**
3:    $x^{k+1} \leftarrow T\left(x^k\right)$
4:    **if** $\left\| x^{k+1} - x^k \right\|_2 < \epsilon$ **then**
5:       return $x^{k+1}$
6:    $k \leftarrow k + 1$

---

## B SPECTRAL STATISTICS AND SPECTRAL ISOMORPHISM MEASURES

We also explored other spectral statistics on monolingual embeddings. The *numeric rank* of $A$ is a smoother analog to rank (where there is noise in low rank components), defined $\eta(A) = \|A\|_F^2 / \|A\|_2^2$. The *condition number* of $A$ is $\kappa(A) = \sigma_1 / \sigma_d$, and measures how close the matrix is to being truly full rank, smaller is more stable. For two matrices $A_1$ and $A_2$, the *condition number harmonic mean* is COND-HM$(A_1, A_2) = \frac{2\kappa(A_1)\kappa(A_2)}{\kappa(A_1)+\kappa(A_2)}$. Smaller means the matrices are more comparable. Figure 3 plots these measures, and again demonstrates that I-C+SN+L improves these measures on matrices.

We also show the raw numbers used to generate the charts in Figure 2 in the tables below.

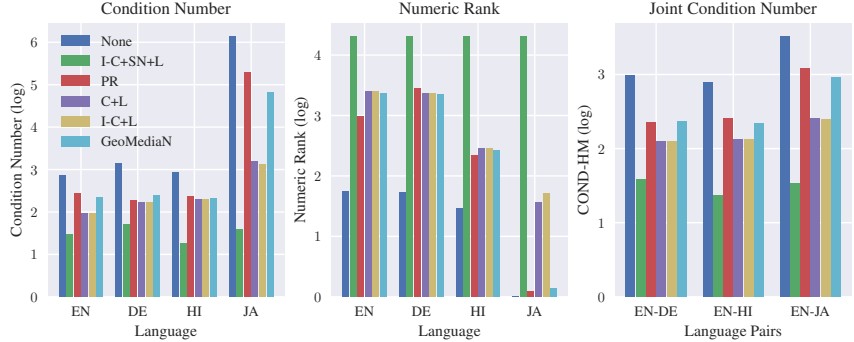

Figure 3: Spectral Measures of four (4) monolingual word embeddings.

Table 8: Effective Rank

| | Normalization Algorithms | | | | | |
|---|---|---|---|---|---|---|
| Languages | None | PR | GeoMediaN | C+L | I-C+L | I-C+SN+L |
| EN | 268 | 277 | 278 | 279 | 279 | 283 |
| DE | 264 | 273 | 273 | 274 | 274 | 278 |
| HI | 258 | 270 | 269 | 269 | 269 | 282 |
| JA | 39 | 96 | 171 | 253 | 255 | 276 |

Table 9: Effective Condition Number

| | Normalization Algorithms | | | | | |
|---|---|---|---|---|---|---|
| Langauges | None | PR | GeoMediaN | C+L | I-C+L | I-C+SN+L |
| EN | 13.1 | 6.7 | 5.4 | 5.3 | 5.3 | 3.3 |
| DE | 14.8 | 5.9 | 6.2 | 6.1 | 6.1 | 3.7 |
| HI | 16.1 | 9.4 | 9.1 | 8.9 | 8.9 | 3.2 |
| JA | 175.0 | 113.3 | 73.1 | 16.4 | 15.0 | 3.4 |

Table 10: Effective Condition Number Harmonic Mean

| Langauges Pairs | Normalization Algorithms | | | | | |
|---|---|---|---|---|---|---|
| | None | PR | GeoMediaN | C+L | I-C+L | I-C+SN+L |
| EN-DE | 13.9 | 6.3 | 5.8 | 5.7 | 5.7 | 3.5 |
| EN-HI | 14.4 | 7.8 | 6.8 | 6.6 | 6.6 | 3.2 |
| EN-JA | 24.4 | 12.7 | 10.0 | 8.0 | 7.8 | 3.3 |

Table 11: Singular Value Gap

| Langauges Pairs | Normalization Algorithms | | | | | |
|---|---|---|---|---|---|---|
| | None | PR | GeoMediaN | C+L | I-C+L | I-C+SN+L |
| EN-DE | 2.3 | 2.4 | 2.2 | 2.2 | 2.2 | 2.2 |
| EN-HI | 49.0 | 48.6 | 12.2 | 12.2 | 12.2 | 5.2 |
| EN-JA | 44.1 | 45.8 | 404.0 | 17.7 | 14.8 | 2.2 |

## C  NORMALIZING CONTEXTUAL TYPE-LEVEL EMBEDDINGS

**Contextual Type-level Embeddings**    To obtain the contextual type-level embeddings, Fast Align (Dyer et al., 2013) is applied to the source-target parallel corpora to derive silver aligned token pairs. The tokenized parallel corpus is fed into pre-trained BERTs (Wolf et al., 2019; Safaya et al., 2020) of the source and target language. Since there will be multiple occurrences of type-level words and each type-level word will possess a contextual word embedding, for each type-level word, its vector representation is derived from the mean vector of all the type-level words from the monolingual corpus fed into the pre-trained language model. This is done for both source and target language. The resulting type-level contextual embedding from the source and target language is then aligned by solving the Procrustes problem.

## D  WORD SIMILARITY TASK

The word similiarity task was conducted using the following English word similarity benchmarks: WS-3533 (Finkelstein et al., 2001), WS-SIM and WSREL (Agirre et al., 2009), RG-65 (Rubenstein & Goodenough, 1965), MC-30 (Miller & Charles, 1991), MTurk-2875 (Radinsky et al., 2011), MTurk-771 (Halawi et al., 2012), MEN7 (Bruni et al., 2012), YP-130 (Yang & Powers, 2006), Rare Words (Luong et al., 2013).

In addition to a baseline (None, which means no normalization), Tabel 12 shows results comparing against our proposed normalization (I-C+SN+L) and the state of the art (I-C+L). Note that both of these improve the accuracy of the similarity tests. This indicates that they are not distorting the critical information contained in the original embeddings. And our proposed approach (I-C+SN+L) increases the score the most.

In contrast, we show the results of two other spectral adjustments we considered. SSV (SSV means Same Singular Values), performs the SVD on the original embedding and then sets all the singular values to $\eta = \sqrt{\|A\|_F^2/d}$. Then we compute U@S@VT to get the new word embedding. So similar to SpecNorm, but it never reaches the else condition. This slightly decreases the scores on the similarity tests. This is a result of accentuating the noise directions. In the other direction, SSVZ

(SSVZ means Set Singular Values to Zero) only keeps the top 40 singular values and then sets the rest to zero. Then we compute U@S@VT to get the new word embedding. This drastically reduces the similarity score. This shows those noise directions, which tend to be well below the average singular value, are still important and cannot just be removed.

Table 12: Monolingual Word Similarity Score (Spearman rank coefficient)

| | Normalization Algorithms | | | | |
|---|---|---|---|---|---|
| Dataset | None | SSV | SSVZ | I-C+L | I-C+SN+L |
| EN_WS-353-ALL | 0.7388 | 0.7127 | 0.5395 | 0.7433 | 0.7555 |
| EN_VERB-143 | 0.3973 | 0.4283 | 0.2635 | 0.4231 | 0.4346 |
| EN_YP-130 | 0.5333 | 0.5534 | 0.3904 | 0.5514 | 0.5631 |
| EN_MTurk-771 | 0.6689 | 0.6583 | 0.5540 | 0.6838 | 0.6926 |
| EN_RG-65 | 0.7974 | 0.7640 | 0.6390 | 0.8082 | 0.8087 |
| EN_RW-STANFORD | 0.5080 | 0.5569 | 0.3873 | 0.5125 | 0.5258 |
| EN_SEMEVAL17 | 0.7216 | 0.7478 | 0.5779 | 0.7288 | 0.7366 |
| EN_MEN-TR-3k | 0.7637 | 0.7506 | 0.6581 | 0.7720 | 0.7792 |
| EN_WS-353-SIM | 0.7811 | 0.7678 | 0.6162 | 0.7897 | 0.7888 |
| EN_MTurk-287 | 0.6773 | 0.6439 | 0.6016 | 0.6864 | 0.6864 |
| EN_WS-353-REL | 0.6820 | 0.6363 | 0.4824 | 0.6905 | 0.7081 |
| EN_MC-30 | 0.8123 | 0.8203 | 0.6754 | 0.8352 | 0.8494 |
| EN_SIMLEX-999 | 0.3823 | 0.4069 | 0.2276 | 0.3899 | 0.3955 |
| Avg | 0.6511 | 0.6498 | 0.5087 | 0.6627 | **0.6711** |

## E    SUMMARY OF MODEL PERFORMANCE AND SIGNIFICANCE TEST

Table 13: Summary of Model Performance on I-C+SN+L vs. No Normalization. Where a significant number of language pairs show an improvement (see Table 14) are in **bold**.

| | Models | | | | | |
|---|---|---|---|---|---|---|
| Dict | CCA | PROC | PROC-B | DLV | RCSLS | VECMAP |
| 1k | **28/28** | **28/28** | **25/28** | **28/28** | 13/28 | |
| 3k | **28/28** | **28/28** | **25/28** | **28/28** | **25/28** | |
| 5k | **28/28** | **28/28** | | **28/28** | **28/28** | |
| – | | | | | | **27/28** |

Table 13 summarizes the performance of I-C+SN+L on several supervised and unsupervised projection-based CLWE models across all the 28 language pairs as presented in Appendix L . After preprocessing the monolingual word embeddings with I-C+SN+L, CCA[SN], PROC[SN] and DLV[SN] outperformed CCA, PROC and DLV respectively on 28 of 28 language pairs across all the translation dictionaries. PROC-B[SN] outperformed PROC-B on 25 of 28 language pairs across 1k and 3k translation dictionaries. The performance of RCSLS[SN] supersedes RCSLS on 25 of 28 language pairs and 28 of 28 language pairs trained on 3k and 5k translation dictionaries respectively. The unsupervised projection-based CLWE model, VECMAP[SN] outperformed VECMAP on 27 of 28 language pairs. The lowest performing model was RCSLS[SN] trained on 1k translation dictionary.

We compare the effectiveness of Iterative Spectral Normalization on $n = 28$ language pairs. We use a 1-tail Binomial Test to measure the significance of the consistency of the improvement. That is, Iterative Spectral Normalization is a significant improvement over no normalization if it improves the score on well more than 50% of the language pairs. Table 14 shows the $p$-value derived from the Binomial test with $n = 28$ language with probability 0.5, for different values $k/n$ language pairs

with an improvement. For instance, $21/28$ language pairs with an improvement have a $p$-value of 0.006 so is significant at the 0.05 level and the 0.01 level. In the associated Tables 13 and 15 we show those results in bold which are significant at the 0.01 level.

Table 14: This table shows the $p$-values corresponding with a 1-tail Binomial Test using $n = 28$ items, $k$ observations, against a rate parameter of 0.5, indicating a null hypothesis where each scenario is equally likely (i.e., that neither provides an improvement).

| k/28 | 18 | 19 | 20 | 21 | 22 | 23 | 24 | 25 | 26 | 27 | 28 |
|------|------|------|------|------|------|------|------|------|------|------|------|
| $p$-value | 0.09 | 0.04 | 0.02 | 0.006 | 0.002 | 0.0004 | 9e-05 | 1e-05 | 2e-06 | 1e-07 | 4e-09 |

Table 15: Comparison of Model Performance of I-C+L vs. I-C+SN+L. The table shows the fraction of language pairs where I-C+SN+L performs better. Where a significant number of language pairs show an improvement (see Table 14) are in **bold**.

| | | | | | Models | |
|------|------|------|------|------|------|------|
| Dict | CCA | PROC | PROC-B | DLV | RCSLS | VECMAP |
| 3k | | | **25/28** | | | |
| 5k | **22/28** | **24/28** | | **23/28** | 14/28 | |
| – | | | | | | **24/28** |

Table 15 summarizes the performance of I-C+L and I-C+SN+L on several supervised and unsupervised projection-based CLWE models across all the 28 language pairs as presented in Appendix K . In all cases, I-C+SN+L outperforms I-C+L except on RCSLS where there was a tie.

Bold signifies that they are significant at a significance level of 0.01 under a Binomial Test, as shown in Table 14.

## F  P@1 PERFORMANCE FOR MAPPING BASED CLWE METHODS VS JOINT TRAINING

Table 16: BLI performance (P@1) on 4 languages from MUSE. $*$ are scores reported from published papers; others are ones we computed.

| Model | Normalization | EN-DE | EN-ES | EN-FR | EN-RU | Avg |
|-------|---------------|-------|-------|-------|-------|-----|
| *Embed then Align Methods* | | | | | | |
| PROC (Conneau et al., 2018a)$*$ | None | 73.5 | 81.4 | 81.1 | 51.7 | 71.9 |
| PROC (Conneau et al., 2018a) | I-C+SN+L | 75.7 | 82.7 | 83.2 | 51.8 | 73.3 |
| RCSLS (Joulin et al., 2018) | None | 43.6 | 50.5 | 53.1 | 26.6 | 43.4 |
| RCSLS (Joulin et al., 2018) | I-C+SN+L | 77.9 | 83.7 | 83.7 | **57.8** | **75.7** |
| ADV (Conneau et al., 2018a)$*$ | None | 74.0 | 81.7 | 82.3 | 44.0 | 70.5 |
| ADV (Conneau et al., 2018a) | I-C+SN+L | 0.00 | 0.00 | 82.9 | 0.07 | 20.7 |
| WP (Grave et al., 2019)$*$ | None | 75.6 | 82.8 | 82.3 | 45.2 | 71.5 |
| WP (Grave et al., 2019) | I-C+SN+L | 77.4 | 82.9 | 83.6 | 47.8 | 72.9 |
| *Joint Training Methods* | | | | | | |
| (Wang et al., 2019)$*$ | None | 74.2 | 81.4 | 82.8 | 45.0 | 70.9 |
| (Ormazabal et al., 2021)$*$ | None | **78.1** | **84.2** | **84.9** | 51.5 | 74.7 |

**Joint training methods.** The CLWE methods proposed by Wang et al. (2019) and Ormazabal et al. (2021) learn a new embedding from scratch including words from both languages. They do so with a different objective function than standard approaches so that it prevents words entirely clustering by language, and so cross-lingual synonyms are encouraged to be embedded close to each other. Wang et al. (2019) accomplished this by encouraging alignment of representations of words. This was enhanced by Ormazabal et al. (2021) via a mechanism (Cross-lingual Anchoring) that extended the skip-gram to have bilingual target embeddings. Ultimately, these can learn a cross-lingual embedding that preserves synonyms, and is optimized over the entire source language data.

**Results.** Both Ormazabal et al. (2021) and Wang et al. (2019) reported P@1 (precision of the 1st nearest neighbor) BLI results for their method – as opposed to the more robust MAP evaluation we use in the remainder of the paper. So we report the BLI performance on the MUSE dataset on 4 languages before and after I-C+SN+L for a variety of alignment techniques in Table 16. We include their published numbers directly for comparison. We also include reported numbers from various other papers, denoted with ∗. The work of Ormazabal et al. (2021) slightly outperforms I-C+SN+L (with RCSLS (Joulin et al., 2018)) on EN-DE, EN-ES, and EN-FR, but under-performs on EN-RU and as a result has a lower overall average.

The result of Ormazabal et al. (2021) are certainly intriguing, and we hope to be able to compare on a broader set of languages and use the MAP evaluation in future work. We remark, however, that such approaches are less flexible than using a post-processing alignment as they require re-training, so it is less clear how to apply for multiple languages, would need to be re-trained for other languages pairs (e.g., DE-FR), and how it will work for lower resource languages where less training data is available.

After applying our proposed I-C+SN+L normalization, ADV (Conneau et al., 2018b) failed to align EN-DE, EN-ES, EN-RU. This method uses a Generative Adversarial Network (GAN), and this phenomenon is explained by Hartmann et al. (2019). In their works, they showed that unit length normalization makes GAN based CLWE method unstable and deteriorates their performance but other supervised alignments are not affected by it. This explains why the performance on PROC (Conneau et al., 2018a), RCSLS (Joulin et al., 2018) and Wasserstein Procrustes (WP) (Grave et al., 2019) are not affected by I-C+SN+L, and in fact, it leads to an improvement in the BLI performance.

## G  ALIGNMENT OF SINGLE-CELL MULTI-OMICS DATA

Table 17: Alignment Performance (FOSCTTM) on Single-cell Sequencing Dataset

| Alignment methods | Normalization | scGEM FOSCTTM | SNAREseq FOSCTTM |
|---|---|---|---|
| *Before Normalization* | | | |
| MMD-MA ∗ | None | 0.2014 | **0.15** |
| UnionCom ∗ | None | 0.296 | 0.265 |
| SCOT ∗ | None | 0.198 | **0.15** |
| *After Normalization* | | | |
| SCOT | I-C+L | 0.201 | 0.395 |
| SCOT | I-C+SN+L | **0.184** | **0.15** |

Single-cell measurements allow scientists to study the various properties of the genome such as gene expression, chromatin accessibility, DNA methylation etc. Therefore there is the need for data integration of these single-cell measurements so that scientists can understand cell development over time and disease. However, due to the lack of cell-to-cell correspondence between the different types of measurements, it makes the process of data integration a challenging task. Motivated by this, (Demetci et al., 2020) proposed a Single-Cell alignment using Optimal Transport (SCOT), an unsupervised learning algorithm that uses Gromov Wasserstein-based optimal transport to recover the cell-to-cell correspondence between two sequencing domains. * represents results taken directly

from (Demetci et al., 2020). MMD-MA (Liu et al., 2019) and UnionCom (Cao et al., 2020) are two other unsupervised single-cell alignment methods.

We follow the implementation details from (Demetci et al., 2020). * represents results taken directly from (Demetci et al., 2020). We use two single-cell multi-omics datasets: 1) scGEM assay (Cheow et al., 2016) and 2) SNAREseq assay (Chen et al., 2019). The fraction of samples is closer than the true match, (FOSCTTM) is used to evaluate the alignment performance. A lower average FOSCTTM score indicates a higher ability of SCOT to recover the correct correspondences.

Table 17 summaries the before and after I-C+L and I-C+SN+L SCOT performance based on FOS-CTTM. On the SNAREseq dataset SCOT with I-C+SN+L outperforms SCOT with I-C+L but achieves the same performance as SCOT without normalization. However, processing the scGEM dataset with I-C+SN+L before alignment achieved the highest FOSCTTM score of 0.184 followed by SCOT without normalization.

## H    CODE FOR SPECNORM

The code implementing our new algorithm Spectral Normalization is provided here https://github.com/poaboagye/SpecNorm. As it is quite simple, for completeness we also present it next.

Spectral Normalization (SpecNorm) is below, referred to as **SpecNorm.py**. Similarly, the code implementation of Iterative Spectral Normalization (I-C+SN+L) referred to as **IterSpecNorm.py** is also shown below.

**Spectral Normalization (SpecNorm.py)**

```python
import numpy as np

def computeSVD(embed):
    """
    Args:
        emded: Monolingual Embedding
    Returns:
        Singular Value Decomposition
    """
    U, S, VT = np.linalg.svd(embed,full_matrices=False)
    return U, S, VT

def specNorm(embed, beta):
    """
    Args:
        emded: Monolingual Embedding
        beta: Use to determine smaller (noisy)
        singular values to be removed
    Returns:
        Spectral Normalized Embedding
    """
    # Perform SVD on the Monolingual Embedding
    _, S, VT = computeSVD(embed)
    # Compute eta
    eta = np.sqrt(np.sum(S**2)/len(S))
    # Transform  diagonal matrix
    S_prime = 1 / S
    for idx, sigma in enumerate(S):
        if sigma > beta*eta:
            S_prime[idx] = S_prime[idx] * (beta*eta)
        else:
            S_prime[idx] = 1
    S_prime = np.eye(len(S)) * S_prime
    # Compute new monolingual embedding
    embed = embed @ VT.T @ S_prime
    return embed
```

**Iterative Spectral Normalization (IterSpecNorm.py)**

```python
import numpy as np
from SpecNorm import specNorm
from argparse import ArgumentParser

def load_embed(filename, max_vocab=-1):
    words, embeds = [], []
    with open(filename, 'r') as f:
        next(f)
        for line in f:
            word, vector = line.rstrip().split(' ', 1)
            vector = np.fromstring(vector, sep=' ')
            words.append(word)
            embeds.append(vector)
            if len(embeds) == max_vocab:
                break
    return words, np.array(embeds)

def saveEmbed(path, words, word_embeds):
    with open(path, 'w') as f:
        print(word_embeds.shape[0], word_embeds.shape[1], file=f)
        for word, embed in zip(words, word_embeds):
            vector_str = ' '.join(str(x) for x in embed)
            print(word, vector_str, file = f)

def main():
    parser = ArgumentParser()
    parser.add_argument('--input_file')
    parser.add_argument('--output_file')
    parser.add_argument('--niter', default=5, type=int)
    parser.add_argument('--max_vocab', default=200000, type=int)
    parser.add_argument('--beta', default=2, type=int)
    args = parser.parse_args()

    words, embeds = load_embed(args.input_file, max_vocab=args.max_vocab)
    embeds /= np.linalg.norm(embeds, axis=1)[:, np.newaxis] + 1e-8

    for i in range(args.niter):
        # Center Monoligual Embedding
        embeds -= embeds.mean(axis=0)[np.newaxis, :]
        # Perform Spectral Normalization
        embeds =  specNorm(embeds, args.beta)
        # Unit Length Normalization
        embeds /= np.linalg.norm(embeds, axis=1)[:, np.newaxis] + 1e-8
    saveEmbed(args.output_file, words, embeds)

if __name__ == '__main__':
    main()
```

## I  RUNTIME

Most of the alignment algorithms run on a CPU except for VecMap, which requires a GPU for faster computation. It takes about 91 seconds to run Iterative Spectral Normalization on a CPU with a $\beta = 2$ and five iterations. Hardware specifications are NVIDIA GeForce GTX Titan Xp 12GB, AMD Ryzen 7 1700 eight-core processor, and 62.8GB RAM. All alignment approaches were completed in under 15 minutes, and most less than 5 minutes. Each evaluation (BLI, CLDC, or XNLI) takes under 2 minutes, but the training step for CLDC and XNLI takes about a day each; hence our approach aims only to need to do this once (on a high resource language like English), and then use the faster alignment step to transfer this to other languages.

Table 18: BLI performance (MAP) on aligning $\mathbf{EN-X}_{L_2}$. We compare all the normalization techniques: None (No normalization), PR (PCA Removal) (Mu & Viswanath, 2018), GeoMediaN (Geometric Median Normalization), C+L (Mean centering and Length normalization, 1 round), I-C+L (Iterative Mean centering and Length normalization, 5 rounds) (Zhang et al., 2019), SN (Spectral Normalization, 1 round), C+SN+L (Mean centering, Spectral Normalization and Length normalization, 1 round), I-C+SN+L (Iterative Mean centering, Spectral Normalization and Length normalization, 5 rounds).

| Method | Normalization | BG | CA | CS | DE | ES | FR | KO | TH | ZH | Avg |
|---|---|---|---|---|---|---|---|---|---|---|---|
| CCA | None | 0.298 | 0.556 | 0.364 | 0.358 | 0.514 | 0.485 | 0.242 | 0.209 | 0.198 | 0.358 |
| | PR | 0.316 | 0.583 | 0.389 | 0.374 | 0.523 | 0.492 | 0.283 | 0.224 | 0.362 | 0.394 |
| | GeoMediaN | 0.316 | 0.580 | 0.383 | 0.376 | 0.524 | 0.492 | 0.277 | 0.226 | 0.362 | 0.393 |
| | C+L | 0.326 | 0.582 | 0.387 | 0.375 | 0.521 | 0.491 | 0.267 | 0.227 | 0.359 | 0.393 |
| | I-C+L | 0.326 | 0.582 | 0.387 | 0.375 | 0.521 | 0.492 | 0.267 | 0.226 | 0.371 | 0.394 |
| | SN | 0.314 | 0.580 | 0.384 | 0.370 | 0.519 | 0.494 | 0.259 | 0.223 | 0.378 | 0.391 |
| | C+SN+L | 0.329 | 0.586 | 0.389 | 0.374 | 0.523 | 0.495 | 0.262 | 0.225 | 0.376 | 0.395 |
| | I-C+SN+L | 0.328 | 0.585 | 0.388 | 0.374 | 0.524 | 0.496 | 0.258 | 0.229 | 0.378 | **0.396** |
| PROC | None | 0.296 | 0.553 | 0.363 | 0.357 | 0.509 | 0.481 | 0.255 | 0.212 | 0.255 | 0.365 |
| | PR | 0.316 | 0.575 | 0.386 | 0.371 | 0.524 | 0.492 | 0.285 | 0.223 | 0.343 | 0.391 |
| | GeoMediaN | 0.317 | 0.578 | 0.384 | 0.376 | 0.521 | 0.491 | 0.281 | 0.225 | 0.346 | 0.391 |
| | C+L | 0.327 | 0.582 | 0.390 | 0.373 | 0.520 | 0.490 | 0.279 | 0.227 | 0.354 | 0.394 |
| | I-C+L | 0.327 | 0.582 | 0.390 | 0.372 | 0.520 | 0.490 | 0.280 | 0.228 | 0.366 | 0.395 |
| | SN | 0.319 | 0.580 | 0.384 | 0.369 | 0.520 | 0.493 | 0.277 | 0.227 | 0.378 | 0.394 |
| | C+SN+L | 0.331 | 0.586 | 0.380 | 0.374 | 0.524 | 0.495 | 0.273 | 0.227 | 0.378 | 0.396 |
| | I-C+SN+L | 0.330 | 0.586 | 0.389 | 0.375 | 0.525 | 0.495 | 0.287 | 0.224 | 0.374 | **0.398** |
| PROC-B | None | 0.326 | 0.587 | 0.400 | 0.382 | 0.528 | 0.497 | 0.236 | 0.218 | 0.221 | 0.377 |
| | PR | 0.340 | 0.605 | 0.425 | 0.395 | 0.536 | 0.505 | 0.259 | 0.227 | 0.341 | 0.404 |
| | GeoMediaN | 0.304 | 0.605 | 0.425 | 0.395 | 0.538 | 0.507 | 0.260 | 0.219 | 0.352 | 0.400 |
| | C+L | 0.354 | 0.607 | 0.423 | 0.396 | 0.536 | 0.507 | 0.257 | 0.223 | 0.366 | 0.408 |
| | I-C+L | 0.354 | 0.608 | 0.424 | 0.396 | 0.536 | 0.508 | 0.257 | 0.224 | 0.380 | 0.410 |
| | SN | 0.347 | 0.602 | 0.421 | 0.392 | 0.533 | 0.504 | 0.257 | 0.229 | 0.389 | 0.408 |
| | C+SN+L | 0.358 | 0.613 | 0.427 | 0.396 | 0.539 | 0.501 | 0.261 | 0.229 | 0.397 | 0.413 |
| | I-C+SN+L | 0.358 | 0.619 | 0.426 | 0.397 | 0.538 | 0.510 | 0.258 | 0.227 | 0.393 | **0.414** |
| RCSLS | None | 0.347 | 0.601 | 0.404 | 0.392 | 0.530 | 0.503 | 0.317 | 0.227 | 0.227 | 0.394 |
| | PR | 0.337 | 0.591 | 0.387 | 0.385 | 0.529 | 0.498 | 0.290 | 0.234 | 0.107 | 0.373 |
| | GeoMediaN | 0.337 | 0.592 | 0.391 | 0.384 | 0.530 | 0.499 | 0.284 | 0.231 | 0.167 | 0.379 |
| | C+L | 0.345 | 0.599 | 0.400 | 0.391 | 0.530 | 0.502 | 0.288 | 0.221 | 0.361 | 0.404 |
| | I-C+L | 0.346 | 0.598 | 0.400 | 0.391 | 0.530 | 0.502 | 0.288 | 0.221 | 0.382 | 0.406 |
| | SN | 0.341 | 0.597 | 0.395 | 0.394 | 0.533 | 0.504 | 0.282 | 0.217 | 0.385 | 0.405 |
| | C+SN+L | 0.348 | 0.601 | 0.403 | 0.393 | 0.533 | 0.506 | 0.285 | 0.215 | 0.377 | **0.407** |
| | I-C+SN+L | 0.348 | 0.601 | 0.401 | 0.392 | 0.533 | 0.506 | 0.280 | 0.214 | 0.376 | 0.406 |

Table 19: BLI performance (MAP) on aligning $\mathbf{X}_{L_1}-\mathbf{EN}$. We compare all the normalization techniques: None (No normalization), PR (PCA Removal) (Mu & Viswanath, 2018), GeoMediaN (Geometric Median Normalization), C+L (Mean centering and Length normalization, 1 round), I-C+L (Iterative Mean centering and Length normalization, 5 rounds) (Zhang et al., 2019), SN (Spectral Normalization, 1 round), C+SN+L (Mean centering, Spectral Normalization and Length normalization, 1 round), I-C+SN+L (Iterative Mean centering, Spectral Normalization and Length normalization, 5 rounds).

| Method | Normalization | BG | CA | CS | DE | ES | FR | KO | TH | ZH | Avg |
|--------|---------------|----|----|----|----|----|----|----|----|----|-----|
| CCA | None | 0.448 | 0.673 | 0.514 | 0.444 | 0.576 | 0.568 | 0.199 | 0.086 | 0.078 | 0.398 |
| | PR | 0.465 | 0.684 | 0.523 | 0.450 | 0.581 | 0.578 | 0.230 | 0.099 | 0.292 | 0.434 |
| | GeoMediaN | 0.467 | 0.688 | 0.523 | 0.449 | 0.582 | 0.583 | 0.231 | 0.098 | 0.279 | 0.433 |
| | C+L | 0.471 | 0.692 | 0.526 | 0.449 | 0.582 | 0.585 | 0.235 | 0.102 | 0.306 | 0.439 |
| | I-C+L | 0.471 | 0.692 | 0.526 | 0.449 | 0.582 | 0.585 | 0.234 | 0.102 | 0.313 | 0.439 |
| | SN | 0.467 | 0.689 | 0.527 | 0.455 | 0.587 | 0.581 | 0.230 | 0.114 | 0.310 | 0.440 |
| | C+SN+L | 0.472 | 0.693 | 0.527 | 0.458 | 0.586 | 0.590 | 0.238 | 0.115 | 0.319 | 0.444 |
| | I-C+SN+L | 0.473 | 0.692 | 0.526 | 0.459 | 0.586 | 0.590 | 0.236 | 0.115 | 0.324 | **0.445** |
| PROC | None | 0.450 | 0.669 | 0.510 | 0.440 | 0.573 | 0.569 | 0.203 | 0.081 | 0.096 | 0.399 |
| | PR | 0.465 | 0.679 | 0.519 | 0.447 | 0.578 | 0.579 | 0.235 | 0.099 | 0.273 | 0.430 |
| | GeoMediaN | 0.468 | 0.685 | 0.519 | 0.449 | 0.581 | 0.582 | 0.236 | 0.100 | 0.267 | 0.432 |
| | C+L | 0.475 | 0.688 | 0.523 | 0.451 | 0.582 | 0.583 | 0.240 | 0.101 | 0.293 | 0.437 |
| | I-C+L | 0.475 | 0.688 | 0.523 | 0.451 | 0.582 | 0.583 | 0.240 | 0.103 | 0.301 | 0.438 |
| | SN | 0.470 | 0.687 | 0.523 | 0.452 | 0.584 | 0.580 | 0.234 | 0.116 | 0.298 | 0.438 |
| | C+SN+L | 0.475 | 0.692 | 0.526 | 0.457 | 0.586 | 0.589 | 0.245 | 0.115 | 0.315 | 0.444 |
| | I-C+SN+L | 0.476 | 0.694 | 0.527 | 0.458 | 0.586 | 0.589 | 0.245 | 0.115 | 0.321 | **0.446** |
| PROC-B | None | 0.453 | 0.675 | 0.531 | 0.458 | 0.576 | 0.579 | 0.211 | 0.077 | 0.085 | 0.405 |
| | PR | 0.477 | 0.693 | 0.546 | 0.465 | 0.585 | 0.587 | 0.253 | 0.110 | 0.261 | 0.442 |
| | GeoMediaN | 0.476 | 0.691 | 0.545 | 0.469 | 0.585 | 0.590 | 0.251 | 0.107 | 0.242 | 0.440 |
| | C+L | 0.483 | 0.698 | 0.550 | 0.468 | 0.584 | 0.590 | 0.259 | 0.111 | 0.264 | 0.445 |
| | I-C+L | 0.483 | 0.698 | 0.550 | 0.469 | 0.583 | 0.590 | 0.255 | 0.113 | 0.290 | 0.448 |
| | SN | 0.479 | 0.697 | 0.553 | 0.470 | 0.588 | 0.590 | 0.251 | 0.133 | 0.302 | 0.451 |
| | C+SN+L | 0.489 | 0.702 | 0.555 | 0.474 | 0.591 | 0.598 | 0.261 | 0.127 | 0.325 | 0.458 |
| | I-C+SN+L | 0.491 | 0.703 | 0.558 | 0.475 | 0.592 | 0.599 | 0.258 | 0.130 | 0.341 | **0.461** |
| RCSLS | None | 0.509 | 0.721 | 0.556 | 0.463 | 0.612 | 0.607 | 0.265 | 0.120 | 0.003 | 0.428 |
| | PR | 0.505 | 0.724 | 0.548 | 0.464 | 0.611 | 0.611 | 0.249 | 0.077 | 0.035 | 0.425 |
| | GeoMediaN | 0.504 | 0.725 | 0.549 | 0.462 | 0.611 | 0.611 | 0.250 | 0.108 | 0.041 | 0.429 |
| | C+L | 0.510 | 0.728 | 0.549 | 0.462 | 0.612 | 0.613 | 0.259 | 0.116 | 0.327 | 0.464 |
| | I-C+L | 0.510 | 0.728 | 0.549 | 0.463 | 0.612 | 0.613 | 0.260 | 0.118 | 0.285 | 0.460 |
| | SN | 0.510 | 0.732 | 0.553 | 0.467 | 0.613 | 0.615 | 0.253 | 0.119 | 0.349 | **0.468** |
| | C+SN+L | 0.505 | 0.729 | 0.549 | 0.466 | 0.612 | 0.610 | 0.251 | 0.118 | 0.354 | 0.466 |
| | I-C+SN+L | 0.505 | 0.727 | 0.549 | 0.466 | 0.612 | 0.610 | 0.251 | 0.118 | 0.352 | 0.466 |

# K  FULL BLI RESULTS FOR ALL 28 LANGUAGE PAIRS, TRANSLATION DICTIONARIES, AND MODELS.

Table 20: BLI performance (MAP) for the first batch (14) of language pairs. We compared the Baseline result from (Glavaš et al., 2019) to I-C+SN+L (denoted SN) and I-C+L result on the BLI task.

| Model | Dict | DE-FI | DE-FR | DE-HR | DE-IT | DE-RU | DE-TR | EN-DE | EN-FI | EN-FR | EN-HR | EN-IT | EN-RU | EN-TR | FI-FR | Avg |
|---|---|---|---|---|---|---|---|---|---|---|---|---|---|---|---|---|
| CCA | 5k | 0.353 | 0.509 | 0.318 | 0.506 | 0.411 | 0.280 | 0.542 | 0.383 | 0.652 | 0.325 | 0.624 | 0.454 | 0.327 | 0.362 | 0.432 |
| CCA$^{\text{I-C+L}}$ | 5k | 0.372 | 0.523 | 0.339 | 0.520 | 0.425 | 0.306 | 0.561 | 0.409 | 0.662 | 0.350 | 0.642 | 0.472 | 0.365 | 0.383 | 0.452 |
| CCA$^{\text{SN}}$ | 5k | 0.371 | 0.528 | 0.340 | 0.527 | 0.426 | 0.303 | 0.568 | 0.410 | 0.665 | 0.356 | 0.648 | 0.476 | 0.372 | 0.387 | **0.455** |
| PROC | 5k | 0.359 | 0.511 | 0.329 | 0.510 | 0.425 | 0.284 | 0.544 | 0.396 | 0.654 | 0.336 | 0.625 | 0.464 | 0.335 | 0.362 | 0.438 |
| PROC$^{\text{I-C+L}}$ | 5k | 0.377 | 0.524 | 0.346 | 0.524 | 0.436 | 0.312 | 0.564 | 0.421 | 0.662 | 0.358 | 0.641 | 0.484 | 0.371 | 0.386 | 0.458 |
| PROC$^{\text{SN}}$ | 5k | 0.378 | 0.531 | 0.350 | 0.531 | 0.440 | 0.312 | 0.570 | 0.421 | 0.670 | 0.366 | 0.650 | 0.490 | 0.380 | 0.388 | **0.463** |
| PROC-B | 3k | 0.362 | 0.514 | 0.324 | 0.508 | 0.413 | 0.278 | 0.532 | 0.380 | 0.642 | 0.336 | 0.612 | 0.449 | 0.328 | 0.350 | 0.431 |
| PROC-B$^{\text{I-C+L}}$ | 3k | 0.355 | 0.527 | 0.335 | 0.516 | 0.371 | 0.290 | 0.539 | 0.419 | 0.651 | 0.356 | 0.622 | 0.429 | 0.356 | 0.372 | 0.438 |
| PROC-B$^{\text{SN}}$ | 3k | 0.359 | 0.535 | 0.342 | 0.524 | 0.378 | 0.293 | 0.545 | 0.415 | 0.657 | 0.362 | 0.631 | 0.443 | 0.368 | 0.376 | **0.445** |
| DLV | 5k | 0.357 | 0.506 | 0.328 | 0.510 | 0.423 | 0.284 | 0.545 | 0.396 | 0.649 | 0.334 | 0.625 | 0.467 | 0.335 | 0.351 | 0.436 |
| DLV$^{\text{I-C+L}}$ | 5k | 0.385 | 0.541 | 0.366 | 0.542 | 0.423 | 0.322 | 0.578 | 0.453 | 0.681 | 0.402 | 0.655 | 0.487 | 0.405 | 0.424 | 0.476 |
| DLV$^{\text{SN}}$ | 5k | 0.384 | 0.549 | 0.365 | 0.548 | 0.424 | 0.326 | 0.582 | 0.449 | 0.684 | 0.404 | 0.661 | 0.488 | 0.407 | 0.431 | **0.479** |
| RCSLS | 5k | 0.395 | 0.536 | 0.359 | 0.529 | 0.458 | 0.324 | 0.580 | 0.438 | 0.675 | 0.375 | 0.652 | 0.510 | 0.386 | 0.395 | 0.472 |
| RCSLS$^{\text{I-C+L}}$ | 5k | 0.396 | 0.576 | 0.370 | 0.549 | 0.479 | 0.341 | 0.645 | 0.471 | 0.712 | 0.428 | 0.679 | 0.557 | 0.452 | 0.421 | **0.505** |
| RCSLS$^{\text{SN}}$ | 5k | 0.404 | 0.569 | 0.370 | 0.550 | 0.480 | 0.345 | 0.636 | 0.465 | 0.713 | 0.419 | 0.687 | 0.557 | 0.439 | 0.416 | 0.504 |
| VECMAP | - | 0.302 | 0.505 | 0.300 | 0.493 | 0.322 | 0.253 | 0.521 | 0.292 | 0.626 | 0.268 | 0.600 | 0.323 | 0.288 | 0.368 | 0.390 |
| VECMAP$^{\text{I-C+L}}$ | - | 0.331 | 0.529 | 0.325 | 0.522 | 0.336 | 0.283 | 0.552 | 0.348 | 0.654 | 0.323 | 0.630 | 0.356 | 0.348 | 0.396 | 0.424 |
| VECMAP$^{\text{SN}}$ | - | 0.343 | 0.539 | 0.326 | 0.533 | 0.337 | 0.293 | 0.559 | 0.355 | 0.660 | 0.333 | 0.635 | 0.368 | 0.352 | 0.400 | **0.431** |

Table 21: BLI performance (MAP) for second batch (14) of language pairs. We compared the Baseline result from (Glavaš et al., 2019) to I-C+SN+L (denoted SN) and I-C+L result on the BLI task.

| Model | Dict | FI-HR | FI-IT | FI-RU | HR-FR | HR-IT | HR-RU | IT-FR | RU-FR | RU-IT | TR-FI | TR-FR | TR-HR | TR-IT | TR-RU | Avg |
|-------|------|-------|-------|-------|-------|-------|-------|-------|-------|-------|-------|-------|-------|-------|-------|-----|
| CCA | 5k | 0.288 | 0.353 | 0.340 | 0.372 | 0.366 | 0.367 | 0.668 | 0.469 | 0.474 | 0.260 | 0.337 | 0.250 | 0.331 | 0.285 | 0.369 |
| CCA$^{I-C+L}$ | 5k | 0.313 | 0.381 | 0.359 | 0.399 | 0.394 | 0.389 | 0.678 | 0.488 | 0.490 | 0.284 | 0.358 | 0.263 | 0.353 | 0.295 | 0.389 |
| CCA$^{SN}$ | 5k | 0.311 | 0.384 | 0.362 | 0.403 | 0.393 | 0.389 | 0.681 | 0.491 | 0.492 | 0.284 | 0.364 | 0.269 | 0.357 | 0.299 | **0.391** |
| PROC | 5k | 0.294 | 0.355 | 0.342 | 0.374 | 0.364 | 0.372 | 0.669 | 0.470 | 0.474 | 0.269 | 0.338 | 0.259 | 0.335 | 0.290 | 0.372 |
| PROC$^{I-C+L}$ | 5k | 0.317 | 0.382 | 0.363 | 0.400 | 0.395 | 0.393 | 0.676 | 0.485 | 0.492 | 0.288 | 0.360 | 0.270 | 0.355 | 0.299 | 0.391 |
| PROC$^{SN}$ | 5k | 0.316 | 0.385 | 0.364 | 0.407 | 0.396 | 0.393 | 0.679 | 0.491 | 0.495 | 0.290 | 0.368 | 0.275 | 0.360 | 0.305 | **0.395** |
| PROC-B | 3k | 0.293 | 0.348 | 0.327 | 0.365 | 0.368 | 0.365 | 0.664 | 0.478 | 0.476 | 0.270 | 0.333 | 0.244 | 0.330 | 0.262 | 0.366 |
| PROC-B$^{I-C+L}$ | 3k | 0.303 | 0.371 | 0.336 | 0.399 | 0.397 | 0.371 | 0.671 | 0.488 | 0.488 | 0.279 | 0.351 | 0.263 | 0.348 | 0.256 | 0.380 |
| PROC-B$^{SN}$ | 3k | 0.303 | 0.374 | 0.337 | 0.403 | 0.399 | 0.377 | 0.678 | 0.488 | 0.491 | 0.286 | 0.360 | 0.267 | 0.356 | 0.264 | **0.384** |
| DLV | 5k | 0.294 | 0.356 | 0.342 | 0.364 | 0.366 | 0.374 | 0.665 | 0.466 | 0.475 | 0.268 | 0.333 | 0.255 | 0.336 | 0.289 | 0.370 |
| DLV$^{I-C+L}$ | 5k | 0.350 | 0.413 | 0.390 | 0.440 | 0.441 | 0.422 | 0.691 | 0.510 | 0.507 | 0.313 | 0.394 | 0.302 | 0.385 | 0.318 | 0.420 |
| DLV$^{SN}$ | 5k | 0.357 | 0.420 | 0.392 | 0.445 | 0.440 | 0.422 | 0.695 | 0.515 | 0.513 | 0.320 | 0.401 | 0.311 | 0.391 | 0.322 | **0.425** |
| RCSLS | 5k | 0.321 | 0.388 | 0.376 | 0.412 | 0.399 | 0.404 | 0.682 | 0.494 | 0.491 | 0.300 | 0.375 | 0.285 | 0.368 | 0.324 | 0.401 |
| RCSLS$^{I-C+L}$ | 5k | 0.334 | 0.404 | 0.384 | 0.431 | 0.413 | 0.424 | 0.701 | 0.519 | 0.496 | 0.296 | 0.401 | 0.299 | 0.382 | 0.330 | 0.415 |
| RCSLS$^{SN}$ | 5k | 0.331 | 0.403 | 0.392 | 0.431 | 0.417 | 0.417 | 0.700 | 0.520 | 0.509 | 0.304 | 0.397 | 0.302 | 0.385 | 0.335 | **0.417** |
| VECMAP | - | 0.280 | 0.355 | 0.312 | 0.402 | 0.389 | 0.376 | 0.667 | 0.463 | 0.463 | 0.246 | 0.341 | 0.223 | 0.332 | 0.200 | 0.361 |
| VECMAP$^{I-C+L}$ | - | 0.301 | 0.388 | 0.363 | 0.433 | 0.434 | 0.403 | 0.684 | 0.492 | 0.485 | 0.258 | 0.367 | 0.249 | 0.359 | 0.219 | 0.388 |
| VECMAP$^{SN}$ | - | 0.289 | 0.398 | 0.350 | 0.438 | 0.431 | 0.407 | 0.689 | 0.497 | 0.487 | 0.270 | 0.386 | 0.251 | 0.365 | 0.194 | **0.389** |

Table 22: BLI performance (MAP) for the first batch (14) of language pairs. We compared the Baseline result from (Glavaš et al., 2019) to I-C+SN+L (denoted SN) result on the BLI task.

| Model | Dict | DE-FI | DE-FR | DE-HR | DE-IT | DE-RU | DE-TR | EN-DE | EN-FI | EN-FR | EN-HR | EN-IT | EN-RU | EN-TR | FI-FR | Avg |
|---|---|---|---|---|---|---|---|---|---|---|---|---|---|---|---|---|
| CCA | 1k | 0.241 | 0.422 | 0.206 | 0.414 | 0.308 | 0.153 | 0.458 | 0.259 | 0.582 | 0.218 | 0.538 | 0.336 | 0.218 | 0.230 | 0.327 |
| CCA$^{SN}$ | 1k | 0.259 | 0.456 | 0.224 | 0.445 | 0.326 | 0.179 | 0.486 | 0.286 | 0.609 | 0.244 | 0.560 | 0.362 | 0.259 | 0.260 | **0.354** |
| CCA | 3k | 0.328 | 0.494 | 0.298 | 0.491 | 0.399 | 0.251 | 0.531 | 0.351 | 0.642 | 0.299 | 0.613 | 0.434 | 0.314 | 0.332 | 0.413 |
| CCA$^{SN}$ | 3k | 0.345 | 0.518 | 0.314 | 0.511 | 0.413 | 0.278 | 0.554 | 0.377 | 0.657 | 0.335 | 0.634 | 0.455 | 0.348 | 0.360 | **0.436** |
| CCA | 5k | 0.353 | 0.509 | 0.318 | 0.506 | 0.411 | 0.280 | 0.542 | 0.383 | 0.652 | 0.325 | 0.624 | 0.454 | 0.327 | 0.362 | 0.432 |
| CCA$^{SN}$ | 5k | 0.371 | 0.528 | 0.340 | 0.527 | 0.426 | 0.303 | 0.568 | 0.410 | 0.665 | 0.356 | 0.648 | 0.476 | 0.372 | 0.387 | **0.455** |
| PROC | 1k | 0.264 | 0.428 | 0.225 | 0.421 | 0.323 | 0.169 | 0.458 | 0.271 | 0.579 | 0.225 | 0.535 | 0.352 | 0.225 | 0.239 | 0.336 |
| PROC$^{SN}$ | 1k | 0.280 | 0.459 | 0.244 | 0.458 | 0.346 | 0.194 | 0.490 | 0.293 | 0.611 | 0.255 | 0.566 | 0.378 | 0.263 | 0.268 | **0.365** |
| PROC | 3k | 0.340 | 0.499 | 0.308 | 0.495 | 0.413 | 0.260 | 0.532 | 0.365 | 0.642 | 0.307 | 0.611 | 0.449 | 0.320 | 0.333 | 0.420 |
| PROC$^{SN}$ | 3k | 0.354 | 0.522 | 0.326 | 0.516 | 0.423 | 0.283 | 0.558 | 0.385 | 0.659 | 0.346 | 0.637 | 0.472 | 0.357 | 0.362 | **0.443** |
| PROC | 5k | 0.359 | 0.511 | 0.329 | 0.510 | 0.425 | 0.284 | 0.544 | 0.396 | 0.654 | 0.336 | 0.625 | 0.464 | 0.335 | 0.362 | 0.438 |
| PROC$^{SN}$ | 5k | 0.378 | 0.531 | 0.350 | 0.531 | 0.440 | 0.312 | 0.570 | 0.421 | 0.670 | 0.366 | 0.650 | 0.490 | 0.380 | 0.388 | **0.463** |
| PROC-B | 1k | 0.354 | 0.511 | 0.306 | 0.507 | 0.392 | 0.250 | 0.521 | 0.360 | 0.633 | 0.296 | 0.605 | 0.419 | 0.301 | 0.329 | 0.413 |
| PROC-B$^{SN}$ | 1k | 0.347 | 0.531 | 0.321 | 0.518 | 0.359 | 0.283 | 0.543 | 0.411 | 0.66 | 0.346 | 0.628 | 0.414 | 0.354 | 0.373 | **0.435** |
| PROC-B | 3k | 0.362 | 0.514 | 0.324 | 0.508 | 0.413 | 0.278 | 0.532 | 0.380 | 0.642 | 0.336 | 0.612 | 0.449 | 0.328 | 0.350 | 0.431 |
| PROC-B$^{SN}$ | 3k | 0.359 | 0.535 | 0.342 | 0.524 | 0.378 | 0.293 | 0.545 | 0.415 | 0.657 | 0.362 | 0.631 | 0.443 | 0.368 | 0.376 | **0.445** |
| DLV | 1k | 0.259 | 0.384 | 0.222 | 0.420 | 0.325 | 0.167 | 0.454 | 0.271 | 0.546 | 0.225 | 0.537 | 0.353 | 0.221 | 0.209 | 0.328 |
| DLV$^{SN}$ | 1k | 0.260 | 0.472 | 0.239 | 0.458 | 0.333 | 0.198 | 0.503 | 0.305 | 0.632 | 0.274 | 0.584 | 0.389 | 0.287 | 0.274 | **0.372** |
| DLV | 3k | 0.341 | 0.496 | 0.306 | 0.494 | 0.411 | 0.261 | 0.533 | 0.365 | 0.636 | 0.307 | 0.611 | 0.444 | 0.320 | 0.321 | 0.418 |
| DLV$^{SN}$ | 3k | 0.361 | 0.540 | 0.339 | 0.537 | 0.414 | 0.300 | 0.571 | 0.418 | 0.677 | 0.381 | 0.651 | 0.471 | 0.393 | 0.399 | **0.461** |
| DLV | 5k | 0.357 | 0.506 | 0.328 | 0.510 | 0.423 | 0.284 | 0.545 | 0.396 | 0.649 | 0.334 | 0.625 | 0.467 | 0.335 | 0.351 | 0.436 |
| DLV$^{SN}$ | 5k | 0.384 | 0.549 | 0.365 | 0.548 | 0.424 | 0.326 | 0.582 | 0.449 | 0.684 | 0.404 | 0.661 | 0.488 | 0.407 | 0.431 | **0.479** |
| RCSLS | 1k | 0.288 | 0.459 | 0.262 | 0.453 | 0.361 | 0.201 | 0.501 | 0.306 | 0.612 | 0.267 | 0.565 | 0.401 | 0.275 | 0.269 | 0.373 |
| RCSLS$^{SN}$ | 1k | 0.282 | 0.465 | 0.247 | 0.459 | 0.347 | 0.197 | 0.508 | 0.305 | 0.635 | 0.266 | 0.577 | 0.403 | 0.273 | 0.271 | **0.374** |
| RCSLS | 3k | 0.373 | 0.524 | 0.337 | 0.518 | 0.442 | 0.296 | 0.568 | 0.404 | 0.665 | 0.357 | 0.637 | 0.491 | 0.364 | 0.367 | 0.453 |
| RCSLS$^{SN}$ | 3k | 0.366 | 0.543 | 0.336 | 0.533 | 0.448 | 0.302 | 0.612 | 0.421 | 0.696 | 0.375 | 0.668 | 0.523 | 0.395 | 0.374 | **0.471** |
| RCSLS | 5k | 0.395 | 0.536 | 0.359 | 0.529 | 0.458 | 0.324 | 0.580 | 0.438 | 0.675 | 0.375 | 0.652 | 0.510 | 0.386 | 0.395 | 0.472 |
| RCSLS$^{SN}$ | 5k | 0.404 | 0.569 | 0.370 | 0.550 | 0.480 | 0.345 | 0.636 | 0.465 | 0.713 | 0.419 | 0.687 | 0.557 | 0.439 | 0.416 | **0.504** |
| VECMAP | - | 0.302 | 0.505 | 0.300 | 0.493 | 0.322 | 0.253 | 0.521 | 0.292 | 0.626 | 0.268 | 0.600 | 0.323 | 0.288 | 0.368 | 0.390 |
| VECMAP$^{SN}$ | - | 0.343 | 0.539 | 0.326 | 0.533 | 0.337 | 0.293 | 0.559 | 0.355 | 0.660 | 0.333 | 0.635 | 0.368 | 0.352 | 0.400 | **0.431** |

Table 23: BLI performance (MAP) for second batch (14) of language pairs. We compared the Baseline result from (Glavaš et al., 2019) to I-C+SN+L (denoted SN) result on the BLI task.

| Model | Dict | FI-HR | FI-IT | FI-RU | HR-FR | HR-IT | HR-RU | IT-FR | RU-FR | RU-IT | TR-FI | TR-FR | TR-HR | TR-IT | TR-RU | Avg |
|---|---|---|---|---|---|---|---|---|---|---|---|---|---|---|---|---|
| CCA | 1k | 0.167 | 0.232 | 0.214 | 0.238 | 0.240 | 0.256 | 0.612 | 0.344 | 0.352 | 0.151 | 0.213 | 0.134 | 0.202 | 0.146 | 0.250 |
| CCA$^{SN}$ | 1k | 0.193 | 0.257 | 0.236 | 0.273 | 0.265 | 0.274 | 0.638 | 0.380 | 0.379 | 0.157 | 0.236 | 0.148 | 0.227 | 0.164 | **0.273** |
| CCA | 3k | 0.264 | 0.328 | 0.306 | 0.346 | 0.345 | 0.348 | 0.659 | 0.452 | 0.449 | 0.232 | 0.308 | 0.211 | 0.309 | 0.252 | 0.343 |
| CCA$^{SN}$ | 3k | 0.289 | 0.359 | 0.331 | 0.375 | 0.377 | 0.366 | 0.672 | 0.476 | 0.469 | 0.257 | 0.332 | 0.240 | 0.329 | 0.269 | **0.367** |
| CCA | 5k | 0.288 | 0.353 | 0.340 | 0.372 | 0.366 | 0.367 | 0.668 | 0.469 | 0.474 | 0.260 | 0.337 | 0.250 | 0.331 | 0.285 | 0.369 |
| CCA$^{SN}$ | 5k | 0.311 | 0.384 | 0.362 | 0.403 | 0.393 | 0.389 | 0.681 | 0.491 | 0.492 | 0.284 | 0.364 | 0.269 | 0.357 | 0.299 | **0.391** |
| PROC | 1k | 0.187 | 0.247 | 0.233 | 0.248 | 0.247 | 0.269 | 0.615 | 0.352 | 0.360 | 0.169 | 0.215 | 0.148 | 0.211 | 0.168 | 0.262 |
| PROC$^{SN}$ | 1k | 0.217 | 0.271 | 0.252 | 0.285 | 0.276 | 0.285 | 0.641 | 0.387 | 0.391 | 0.178 | 0.243 | 0.166 | 0.239 | 0.182 | **0.287** |
| PROC | 3k | 0.269 | 0.328 | 0.310 | 0.346 | 0.350 | 0.353 | 0.659 | 0.455 | 0.455 | 0.241 | 0.312 | 0.219 | 0.312 | 0.262 | 0.348 |
| PROC$^{SN}$ | 3k | 0.296 | 0.365 | 0.337 | 0.381 | 0.384 | 0.371 | 0.671 | 0.474 | 0.472 | 0.262 | 0.336 | 0.248 | 0.336 | 0.279 | **0.372** |
| PROC | 5k | 0.294 | 0.355 | 0.342 | 0.374 | 0.364 | 0.372 | 0.669 | 0.470 | 0.474 | 0.269 | 0.338 | 0.259 | 0.335 | 0.290 | 0.372 |
| PROC$^{SN}$ | 5k | 0.316 | 0.385 | 0.364 | 0.407 | 0.396 | 0.393 | 0.679 | 0.491 | 0.495 | 0.290 | 0.368 | 0.275 | 0.360 | 0.305 | **0.395** |
| PROC-B | 1k | 0.263 | 0.328 | 0.315 | 0.335 | 0.343 | 0.348 | 0.665 | 0.467 | 0.466 | 0.247 | 0.305 | 0.210 | 0.298 | 0.230 | 0.344 |
| PROC-B$^{SN}$ | 1k | 0.296 | 0.365 | 0.337 | 0.408 | 0.392 | 0.371 | 0.678 | 0.486 | 0.483 | 0.280 | 0.357 | 0.255 | 0.346 | 0.246 | **0.379** |
| PROC-B | 3k | 0.293 | 0.348 | 0.327 | 0.365 | 0.368 | 0.365 | 0.664 | 0.478 | 0.476 | 0.270 | 0.333 | 0.244 | 0.330 | 0.262 | 0.366 |
| PROC-B$^{SN}$ | 3k | 0.303 | 0.374 | 0.337 | 0.403 | 0.399 | 0.377 | 0.678 | 0.488 | 0.491 | 0.286 | 0.360 | 0.267 | 0.356 | 0.264 | **0.384** |
| DLV | 1k | 0.184 | 0.244 | 0.225 | 0.214 | 0.245 | 0.264 | 0.585 | 0.320 | 0.358 | 0.161 | 0.194 | 0.144 | 0.209 | 0.161 | 0.251 |
| DLV$^{SN}$ | 1k | 0.217 | 0.275 | 0.249 | 0.290 | 0.286 | 0.286 | 0.645 | 0.398 | 0.393 | 0.174 | 0.266 | 0.164 | 0.252 | 0.182 | **0.291** |
| DLV | 3k | 0.269 | 0.331 | 0.307 | 0.331 | 0.348 | 0.353 | 0.653 | 0.446 | 0.452 | 0.243 | 0.306 | 0.219 | 0.311 | 0.261 | 0.345 |
| DLV$^{SN}$ | 3k | 0.324 | 0.390 | 0.364 | 0.417 | 0.415 | 0.394 | 0.684 | 0.495 | 0.495 | 0.294 | 0.373 | 0.276 | 0.361 | 0.288 | **0.398** |
| DLV | 5k | 0.294 | 0.356 | 0.342 | 0.364 | 0.366 | 0.374 | 0.665 | 0.466 | 0.475 | 0.268 | 0.333 | 0.255 | 0.336 | 0.289 | 0.370 |
| DLV$^{SN}$ | 5k | 0.357 | 0.420 | 0.392 | 0.445 | 0.440 | 0.422 | 0.695 | 0.515 | 0.513 | 0.320 | 0.401 | 0.311 | 0.391 | 0.322 | **0.425** |
| RCSLS | 1k | 0.214 | 0.272 | 0.257 | 0.281 | 0.275 | 0.291 | 0.637 | 0.381 | 0.383 | 0.194 | 0.247 | 0.170 | 0.246 | 0.191 | **0.289** |
| RCSLS$^{SN}$ | 1k | 0.217 | 0.271 | 0.253 | 0.284 | 0.279 | 0.286 | 0.645 | 0.388 | 0.393 | 0.179 | 0.243 | 0.168 | 0.239 | 0.185 | 0.288 |
| RCSLS | 3k | 0.296 | 0.362 | 0.341 | 0.384 | 0.382 | 0.379 | 0.673 | 0.477 | 0.472 | 0.272 | 0.348 | 0.256 | 0.340 | 0.290 | 0.377 |
| RCSLS$^{SN}$ | 3k | 0.301 | 0.372 | 0.345 | 0.392 | 0.388 | 0.382 | 0.684 | 0.489 | 0.482 | 0.270 | 0.363 | 0.259 | 0.348 | 0.291 | **0.383** |
| RCSLS | 5k | 0.321 | 0.388 | 0.376 | 0.412 | 0.399 | 0.404 | 0.682 | 0.494 | 0.491 | 0.300 | 0.375 | 0.285 | 0.368 | 0.324 | 0.401 |
| RCSLS$^{SN}$ | 5k | 0.331 | 0.403 | 0.392 | 0.431 | 0.417 | 0.417 | 0.700 | 0.520 | 0.509 | 0.304 | 0.397 | 0.302 | 0.385 | 0.335 | **0.417** |
| VECMAP | - | 0.280 | 0.355 | 0.312 | 0.402 | 0.389 | 0.376 | 0.667 | 0.463 | 0.463 | 0.246 | 0.341 | 0.223 | 0.332 | 0.200 | 0.361 |
| VECMAP$^{SN}$ | - | 0.289 | 0.398 | 0.350 | 0.438 | 0.431 | 0.407 | 0.689 | 0.497 | 0.487 | 0.270 | 0.386 | 0.251 | 0.365 | 0.194 | **0.389** |