# OpenReview forum: "Normalization of Language Embeddings for Cross-Lingual Alignment"
_ICLR.cc/2022/Conference — ICLR 2022 Poster_

### Official Review · Reviewer_Qc8y · 2021-10-28

**Correctness:** 4
**Technical Novelty And Significance:** 3
**Empirical Novelty And Significance:** 3
**Recommendation:** 8
**Confidence:** 4

**Details Of Ethics Concerns:**

I have no concerns.

**Main Review:**

Strengths
- Clear description of background knowledge and motivations needed to understand the proposed representation model.
- Clear exposition of the proposed method.
- The authors perform a comprehensive comparison across different benchmarks and downstream tasks.

Questions to the Authors. Please address the following questions during the rebuttal:

- Please elaborate on how cross validation could affect the results in the cases where the scores between methods are close.
- Could you elaborate on the set-up for selecting the hyper-parameters for the proposed model? Given that is based on cross validation, how the model would behave on a low resource setting?
- Could a visualization of the learned representations be useful for the word similarity embedding tasks? Comparing with and without normalisation.
- A possible extra contribution could be to highlight  the single-cell task results on main text.


**Summary Of The Paper:**

The authors propose a normalisation method for cross-lingual text representations.  The goal is to normalise the monolingual embeddings based on spectral normalisation.  The main contributions are: novel method to normalise word embeddings, the proposed method includes different normalization approaches, and the proposed method improves performance on intrinsic and extrinsic evaluation tasks. The study shows that produced text representations keep their meaning and improve performance on downstream tasks.


**Summary Of The Review:**

I recommend acceptance given how clear the paper describes related work, motivations, and proposed model. The authors perform an exhaustive evaluation of the proposed model with different language downstream tasks. Moreover, the model outperforms related work on the downstream tasks.

---

> ### Author Response · Authors · 2021-11-19
> **Response to Reviewer Qc8y**
>
> ***Questions to the Authors. Please address the following questions during the rebuttal:***
>
> -   Please elaborate on how cross-validation could affect the results in the cases where the scores between methods are close.
>
> ***RESPONSE:***
>
> The key reason we employed cross-validation in setting the hyper-parameters is that we wanted to demonstrate that we were not somehow over-fitting to the languages used for evaluation.  The fact that the scores between various parameter settings were similar also indicates that the method does not depend on intricately tuning these values.
>
> This also allowed us to demonstrate that these methods retained performance on standard monolingual tasks, that is it does not show to greatly distort the structure of the data being represented at the expense of a good alignment.
>
>
> -   Could you elaborate on the set-up for selecting the hyper-parameters for the proposed model?
>
> ***RESPONSE:***
>
> **Hyperparameter Tuning, Table 1&2:**
> To avoid overfitting, we performed cross-validation on a held-out set of **5 languages** English (EN), Hindi (HI), Russian (RU), Chinese (ZH), Japanese (JA), Turkish (TR). **Ten (10) Language pairs** of the form EN-X and X-EN were used for hyperparameter tuning for iterative spectral normalization algorithm. We used the publicly available **MUSE translation dictionary** for hyperparameter tuning. Procrustes with the refinement method were used to learn a transfer function trained on 5k source words and evaluated on 1.5k source test queries. Since  $\beta$ could take on 5 values each and with \#Iter $\in $ {1,2,3,4,5} we had a combination of 25  ($\beta$, \#Iter ) pairs each, and out of which we had to choose the best hyperparameters. The best hyperparameter was chosen based on the best average MAP on all 10 language pairs for a given pair and ($\beta$, \#Iter ) pair.
>
> -   Given that is based on cross-validation, how the model would behave in a low-resource setting?
>
> ***RESPONSE:***
>
> We believe we can take these hyper-parameter settings and use them in other tasks, without re-tuning them.  It appears the scores are fairly stable, and the settings are intuitive.  So if a user wanted to employ this on a new setting, and did not have the luxury of other pairs of embeddings to tune parameters on, then we will suggest they use the same ones as in the evaluation tasks in the paper.
>
>
> -   Could a visualization of the learned representations be useful for the word similarity embedding tasks? Comparing with and without normalization.
>
> ***RESPONSE:***
>
> We agree visualization of representations of word embeddings and other embeddings could potentially be useful.  While there are some methods in published work, we understand this is still an active research area.
>
> We have added a figure (Figure 1) illustrating a toy example of the sort of advantage Spectral Normalization could bring to these challenges.
>
>
> -   A possible extra contribution could be to highlight the single-cell task results on the main text.
>
> ***RESPONSE:***
>
> We wish we had space to discuss this in more detail, but are not sure what else to cut.

---

> > ### Comment · Reviewer_Qc8y · 2021-11-29
> > **Questions Addressed**
> >
> > Thank you for addressing my comments. I have no further questions.

---

> ### Author Response · Authors · 2021-11-19
> **General Comments**
>
>
> We thank the reviewers for their insightful comments.  Sorry for the slow reply, we ran a significant number of new experiments, based on your comments, which took time to complete.
> Towards addressing the comments we have updated the paper in three significant ways.  (As a result, some other parts were trimmed.)
>
>
> 1.  We inserted an example (Figure 1, which appears in Section 3) which attempts to illustrate some of the intuition behind how Spectral Normalization can help with alignment, in ways the verbal description in the paper seems to have failed to adequately convey.  It shows a toy example in 2d with corresponding words in English and Japanese.  After centering and length normalization, all words live on the unit circle.  Despite the same ordering of words, the ones in Japanese have ones clustered at the top and bottom of the unit circle, while in English they are clustered on the left and right.  This prevents any great orthogonal alignment.  However, with Spectral Normalization also applied, the words cannot be clustered in the same way, and are made more spread out, while still on the unit circle.  This allows for a much better alignment.
> While the effects of Spectral Normalization in practice are quite likely much more subtle when considered in higher dimensions and on the full data sets (although the spectral properties of English and Japanese are extremely different, so perhaps not), we hope this provides the missing intuition about how Spectral Normalization could be helping.
>
>
> 2.  We added several experiments using the I-C+L normalization as a baseline, this included in the contextual embedding comparison (Table 5), the Cross-lingual document classification task (Table 6), the cross-lingual natural language inference task (Table 7), Alignment of Single-cell Multi-omics Data (Table 17 in Appendix G), and BLI performance comparison on all the 28 language pairs in the GTrans dictionary (Tables 20, 21 in Appendix K, and their summary results in Table 15 in Appendix E).  Similar to the results on the held-out language pairs (Table 3), the I-C+SN+L normalization mostly out-performs I-C+L normalization, but not as much the no normalization baseline.  There are a few exceptions, notable RCSLS which learns an affine transform in addition to the rigid transformation; although in most cases I-C+SN+L still performs the best.  In some cases, the non-rigid transform learned by RCSLS may be inappropriate, or not as effective as in the XNLI task.  Overall, these extensive additional experiments with the additional I-C+L baseline further confirm the helpfulness of the Spectral Normalization.
>
>
> 3.  We attempted to quantify the significance of the improvement gained by Spectral Normalization.  While the numerical improvement of including this approach in addition to centering and length normalization is often small, it is quite consistent.  Measuring a statistical significance on any one task is difficult also since the techniques and experiments are deterministic.  However, we did show significance in the consistency of the improvement using a Binomial test.  This is shown in Appendix E.  For instance, in Table 15, using PROC-B alignment, I-C+SN+L normalization improves over I-C+L normalization on 25 of 28 languages.  Under the Binomial test, this has a p-value of $0.00001$, and is significant at reasonable significance levels (e.g., 0.01).

---

### Official Review · Reviewer_vFnV · 2021-11-01

**Correctness:** 4
**Technical Novelty And Significance:** 3
**Empirical Novelty And Significance:** 2
**Recommendation:** 6
**Confidence:** 4

**Main Review:**

Strengths:
- The proposed method is simple to implement.
- The experiments are extensive and confirms that the proposed normalization method consistently improves cross-lingual word embeddings.

Weaknesses:
- The paper may have limited impact. While the paper does a good job improving alignment-based CLWE, these models are outperformed by multilingual transformers (e.g., multilingual BERT and XLM-R) in many applications. For example, the XNLI results reported in the paper are much worse than that from XLM-R. Therefore, it is unclear if the proposed method is useful for practitioners. One way to improve the impact is extending the method to contextualized embeddings, but this is non-trivial.
- The intuition behind the method is not immediately obvious. The paper may be clearer if there are more intuitive explanations of why the spectral statistics in section 2.2 helps alignment. It is also helpful to have a more explicit discussion of why the proposed method improves the spectral properties from section 2.2.

**Summary Of The Paper:**

This paper studies learning cross-lingual word embeddings (CLWE) with alignment: mapping two pre-trained monolingual word embeddings to a shared space with a linear/orthogonal mapping. Previous work shows that normalization methods and the spectral properties of monolingual word embeddings have a big impact on alignment methods. Building on these results, the paper proposes a normalization method that regularizes the spectral properties of monolingual word embeddings. The method is then combined with mean centering and length normalization in an iterative process. Empirically, the proposed normalization method improves alignment and leads to better CLWE (measured by scores on bilingual lexicon induction, cross-lingual document classification, and natural language inference).

**Summary Of The Review:**

The paper proposes a simple method for improving CLWE with good empirical results. However, the impact may be limited as multilingual transformers are replacing CLWE in many applications. Therefore, my recommendation is weak accept.

---

> ### Author Response · Authors · 2021-11-19
> **Response to Reviewer vFnV**
>
> ***C1:***
>
> The paper may have a limited impact. While the paper does a good job improving alignment-based CLWE, these models are outperformed by multilingual transformers (e.g., multilingual BERT and XLM-R) in many applications. For example, the XNLI results reported in the paper are much worse than those from XLM-R. Therefore, it is unclear if the proposed method is useful for practitioners. One way to improve the impact is extending the method to contextualized embeddings, but this is non-trivial.
>
> ***RESPONSE:***
>
> The goal of this paper is to focus on the alignment of fixed embedded representation.  We focus on language data, and cross-lingual alignment because that is where the most complete data is, and the most well-studied problems.  Given the rise of foundation models (c.f. https://arxiv.org/pdf/2108.07258.pdf), many of which, outside of language, will be and are non-contextual embeddings, we feel this task stands by itself.  For instance, it is not clear how to add context into an image embedding (outside, perhaps, for video).  For instance, we do demonstrate the effectiveness of our spectral normalization on an application to Biology where there is no notion of contextual embeddings.
>
> We did compare against a jointly learned embedding or Ormazabal et al (2021)  and Wang et al (2019), where we could ensure the data sets and evaluation was precisely the same, and on these P@1 evaluations (not as robust as MRR evaluations we do elsewhere), the non-contextual embeddings with I-C+SN+L normalization perform similarly, and better on average.
>
> We also show how to integrate Iterative Spectral Normalization into a very recent paper by Xu & Koehn which aligns contextual embeddings.  This can be found in Table 5.  It shows that I-C+SN+L has consistent improvement over both No normalization and also I-C+L normalization.
>
>
>
> ***C2:***
> The intuition behind the method is not immediately obvious. The paper may be clearer if there are more intuitive explanations of why the spectral statistics in section 2.2 helps alignment. It is also helpful to have a more explicit discussion of why the proposed method improves the spectral properties from section 2.2.
>
> ***RESPONSE:***
>
> As discussed above, we added a new Figure 1 that attempts to illustrate how Spectral Normalization can help on a simple example.

---

> ### Author Response · Authors · 2021-11-19
> **General Comments**
>
>
> We thank the reviewers for their insightful comments.  Sorry for the slow reply, we ran a significant number of new experiments, based on your comments, which took time to complete.
> Towards addressing the comments we have updated the paper in three significant ways.  (As a result, some other parts were trimmed.)
>
>
> 1.  We inserted an example (Figure 1, which appears in Section 3) which attempts to illustrate some of the intuition behind how Spectral Normalization can help with alignment, in ways the verbal description in the paper seems to have failed to adequately convey.  It shows a toy example in 2d with corresponding words in English and Japanese.  After centering and length normalization, all words live on the unit circle.  Despite the same ordering of words, the ones in Japanese have ones clustered at the top and bottom of the unit circle, while in English they are clustered on the left and right.  This prevents any great orthogonal alignment.  However, with Spectral Normalization also applied, the words cannot be clustered in the same way, and are made more spread out, while still on the unit circle.  This allows for a much better alignment.
> While the effects of Spectral Normalization in practice are quite likely much more subtle when considered in higher dimensions and on the full data sets (although the spectral properties of English and Japanese are extremely different, so perhaps not), we hope this provides the missing intuition about how Spectral Normalization could be helping.
>
>
> 2.  We added several experiments using the I-C+L normalization as a baseline, this included in the contextual embedding comparison (Table 5), the Cross-lingual document classification task (Table 6), the cross-lingual natural language inference task (Table 7), Alignment of Single-cell Multi-omics Data (Table 17 in Appendix G), and BLI performance comparison on all the 28 language pairs in the GTrans dictionary (Tables 20, 21 in Appendix K, and their summary results in Table 15 in Appendix E).  Similar to the results on the held-out language pairs (Table 3), the I-C+SN+L normalization mostly out-performs I-C+L normalization, but not as much the no normalization baseline.  There are a few exceptions, notable RCSLS which learns an affine transform in addition to the rigid transformation; although in most cases I-C+SN+L still performs the best.  In some cases, the non-rigid transform learned by RCSLS may be inappropriate, or not as effective as in the XNLI task.  Overall, these extensive additional experiments with the additional I-C+L baseline further confirm the helpfulness of the Spectral Normalization.
>
>
> 3.  We attempted to quantify the significance of the improvement gained by Spectral Normalization.  While the numerical improvement of including this approach in addition to centering and length normalization is often small, it is quite consistent.  Measuring a statistical significance on any one task is difficult also since the techniques and experiments are deterministic.  However, we did show significance in the consistency of the improvement using a Binomial test.  This is shown in Appendix E.  For instance, in Table 15, using PROC-B alignment, I-C+SN+L normalization improves over I-C+L normalization on 25 of 28 languages.  Under the Binomial test, this has a p-value of $0.00001$, and is significant at reasonable significance levels (e.g., 0.01).

---

### Official Review · Reviewer_zhG1 · 2021-11-02

**Correctness:** 2
**Technical Novelty And Significance:** 2
**Empirical Novelty And Significance:** 3
**Recommendation:** 5
**Confidence:** 4

**Main Review:**

### Strengths

The simplicity of the approach and combination with existing preprocessing approaches, is promising as a utility for future work.

Empirical improvement on several extrinsic tasks across multiple domains identify a benefit to SN. Reasonable spectral property-based metrics are shown to improve (in the appropriate direction) when using this method as opposed to methods that do not consider such properties.

The breadth of domains of study is commendable as the authors have clearly taken great care to illustrate that their method has wide applicability. The appendices show further breadth and the argument for the utility of the method is strongly presented.

### Weaknesses

This method shows merit for the reasons given above, but I can describe my primary concern as this — you have proposed solution X for some method A, claiming that A is now improved due to a reduction in effect Z. But you do not demonstrably support that solution X influences effect Z or highlight if effect Z is a critical failure in A. Therefore, while empirical results are strong, it remains unclear what solution X does to improve A. Even if the influence of X on A is not well understood -- this should be delineated further to support that the empirical claims require further introspection.

The work does not make a self-contained case for what  spectral normalization is a solution towards. The problem is poorly defined, which leads to a gap in understanding the motivating strategy of the work. For example, in Section 3 the work states “As a result, if the spectral properties are extreme, it can help regularize them.” -- but it is not clear what it means to (i) regularize such properties or (ii) what this intends to do with respect to the embedding spaces. This could be improved if the problem hypothesis was clearer e.g. answering questions such as:

- Are spectral properties of monolingual embedding spaces malformed?

- Can this be demonstrably proved in this work so the problem to be solved is clear?

- Are there confounding examples within embedding spaces to motivate this (e.g. unexpected versions of the common king-queen=man-woman analogies based on low-frequency words)?

- What does spectral normalisation hope to intrinsically accomplish such that the findings can be better interpreted?

Care is not taken to explicitly link together the spectral statistics to additional methods or to explain either what spectral metrics are informing the reader of in 2.2. It is somewhat of a foregone conclusion that spectrum-based metrics will improve with a spectrum-based method. While this section relies on the insight of Dubossarsky et al. (2020), there is little effort in interrogating the prior work and motivating why prior insight is directly applicable here as-is. At present, this must be taken at face value with little motivation for the succeeding sections. This caveat is somewhat improved in Appendix B, but I consider this insufficient without a clear hypothesis tying the motivation of this approach together.

Overall, there is a lack of research questions in the work which leads to a difficulty in scoping the contribution. SN is clearly useful -- but why? We observe empirical benefits to several tasks, but the work does not source these benefits to specific components or processes within the method. This could be improved with intrinsic analysis (and some error analysis) identifying the effect of SN. This is hinted at in Section 4 where it is stated that “We hypothesize this is because it is somewhat uniformly stretching words along the major modes of variation” however this could be expanded to aid in understanding of the mechanics of the method.

### Unsupported claims

Every claim should be supported with evidence in literature or in the work. There are several instances of claims without proper argument.

“Yet, previous work has clearly demonstrated that there exists significant overall structural similarity, and alignment seeks to make correspondences between those structures for translation and joint understanding.” - I would not consider the arguments of prior work as sufficient for motivation without a in-depth of explanation in this work. This work is not an appendix on prior publication.

“effectively removes much of the clumping of words based on properties other than the pairwise similarity which encodes meaning.” - this term is not introduced or explained as a consequence or issue within CLWE prior art. It is unclear what this means, or how it is required to be mitigated for better CLWE.

“This buttresses the claim made by Xu & Koehn (2021) that improving the degree of isomorphism” - it is unclear where you demonstrate that you improve isomorphism with your method. If your iterative method subsumes methods already maintaining/improving isomorphism then it would be beneficial to demonstrate that such properties are maintained. If this is intended to be obvious, please state so.

“Our approach generalizes previous approaches” -- this is not proven (see previous comment), and little care is taken to show that SN does not interfere with complementary components like Length Normalization.


#### Questions
- In 2.1: what does “longer vectors” refer to here when you are explicitly working with fixed-length vectors? It could be clearer when reduction methods like PCA are cutting up embedding spaces and when frequency methods are used over others.

- What is the difference between matrix X in 2 and A in 2.2? It is unclear why these are re-defined.

- Section 3 -- what is ‘critical information’ in reference to? It appears some assumption is made here, but this is unclear.

- Was a stopping condition explored in the iterative algorithm as opposed to a fixed number of iterations?

- It is stated that the vocabularies are trimmed to the top 200K words, but it would be interesting to see how SN-based methods influence the long tail of infrequent vocabulary. Was this considered?

- Were multi-lingual (L>2) alignments considered?

Please address the unsupported claims and issues raised earlier:
- Is Isomorphism maintained and can you verify that your method subsumes prior methods with absolute certainty?
- What is meant by clumping?
- What are the motivating failings of prior work that has led to the SN method?
- What is the intended intuition and understanding of “regularizing an embedding space”?



#### Typos/Style (not intended as criticism)
- Page 3: why state "wlog" here?

- SN references both the iterative method and the SN method alone without clear discussion of the difference so should be more clearly separated when you are referring to each method.

- Equation numbering gets lost in appendices

- There is repeated usage of embedded clauses without clear demarcation e.g. “This in some sense allows” -> “This, in some sense, allows”. This is generally more of a spoken pattern and could be omitted entirely.

- Procrustes alignment based methods are inconsistently referenced as a “problem”, “solution” or “alignment” with additional unclear italicization. These could be streamlined for consistency.

- Imprecision in technical descriptions -> “terminate this iterative process after a few steps”. What is few here? This is additionally challenging in Section 4 “Alignment Algorithms”

- Inline equation labeling for 2.1-PCA Removal for “d” is lost.

- Figure 1 -- every key in the figure should be explained in the caption. It is not a given to recall what “I-C+L” means in contrast to “C+L”.

- Inconsistent spelling of “Normalization”

**Summary Of The Paper:**

The paper proposes an Iterative Spectral Normalization method to improve pairwise Cross-Lingual Word Embedding alignment. The work supposes that improvement to spectral properties in monolingual embedding spaces are sufficient to yield improvements to Bi-lingual Lexicon Induction, Cross-Lingual Document Classification and Cross-Lingual NLI. The main contributions are (a) the introduction of a simple, portable algorithm to improve pre-processing of embedding spaces for cross-lingual alignment and (b) empirical improvements in multiple domains of cross-lingual embedding oriented tasks.


**Summary Of The Review:**

The recommendation is based on the lack of introspection into the method itself rather than the empirical improvement. The work could be improved if it was better understood what the method is doing and how this ties into performance improvements.

I believe the direction of this work is novel, but overall there is a lack of appropriate setting which ultimately makes it unclear what problem SN solves, and how this fits into existing pre-processing approaches. The case for SN is not fluently well-motivated and there is little analysis to confidently support that SN is superior to prior art or generalises existing approaches.

I support the merit of this method with empirical utility and wide applicability, however, I recommend that significant further revisions are required prior to publication in a later venue.

---

> ### Author Response · Authors · 2021-11-19
> **Response to Reviewer zhG1**
>
>  ***C6: Questions (Part 1)***
>
> -   In 2.1: what does “longer vectors” refer to here when you are explicitly working with fixed-length vectors? It could be clearer when reduction methods like PCA are cutting up embedding spaces and when frequency methods are used over others.
>
> ***RESPONSE:***
>
> The vectors are only unit length after length normalization.  So before that, on input, some are longer than others.  Prior work has shown this is tied to language- or corpus-specific properties like work frequency.
>
>
>  ***C6: Questions (Part 2)***
>
> -   What is the difference between matrix X in 2 and A in 2.2? It is unclear why these are re-defined.
>
>     ***RESPONSE:***
>
> We have changed all notation to be A.  They were explaining different aspects of the algorithms used, but are similar enough it makes sense to use a common notation, hence the change.
>
>  ***C6: Questions (Part 3)***
>
> -   Section 3 -- what is ‘critical information in reference to? It appears some assumption is made here, but this is unclear.
>
> ***RESPONSE:***
>
> Critical information was referring to the similarity between words.
> In some exploratory analysis on the way to develop our proposed Spectral normalization we attempted to set all of the smaller singular values to 0.  This led to very poor performance in all of the evaluation tasks: both cross-lingual as well as monolingual.
> We believe similar observations have been reported elsewhere in attempts to reduce the dimensionality (e.g., with PCA) of the word embeddings, that resulted in poor performance.
>
>
>  ***C6: Questions (Part 4)***
>
> -   Was a stopping condition explored in the iterative algorithm as opposed to a fixed number of iterations?
>
>   ***RESPONSE:***
>
> We considered it, but it seemed more complicated and arbitrary to design a stopping condition that applied to a broad array of settings.  Under cross-validation we found that the result was not sensitive to the number of iterations, and the optimal number was small.  Given this simple solution, we did not feel the need to deeply pursue something else more complicated.
>
>
>  ***C6: Questions (Part 5)***
>
> -   It is stated that the vocabularies are trimmed to the top 200K words, but it would be interesting to see how SN-based methods influence the long tail of infrequent vocabulary. Was this considered?
>
>   ***RESPONSE:***
>
> No, we did not strongly consider an even longer tail.  This is the largest set of words commonly used in the dozens of papers exploring this topic.  For most languages, the total number of words is under 300K, and as one gets even further in the tail, the words become quite obscure.
>
>
>  ***C6: Questions (Part 6)***
>
>
> -   Were multi-lingual (L>2) alignments considered?
>
> ***RESPONSE:***
>
> No.  A simple solution would be to align all languages onto the highest-resource one (English), but other approaches are possible, and the design space for this challenge is broad.  And we felt that fully and fairly exploring it is beyond the scope of the current paper.

---

> ### Author Response · Authors · 2021-11-19
> **Response to Reviewer zhG1**
>
> ***C5: Unsupported claims  (Part 1)***
>
> 1.  “Yet, previous work has clearly demonstrated that there exists significant overall structural similarity, and alignment seeks to make correspondences between those structures for translation and joint understanding.” - I would not consider the arguments of prior work as sufficient for motivation without an in-depth of explanation in this work. This work is not an appendix on prior publication.
>
> ***RESPONSE:***
>
> There are dozens of papers demonstrating the effectiveness of cross-lingual alignment.  We are referring to the implication that this process should only work if there is some inherent common structure among languages that these embeddings are capturing.
>
> More specific aspects of structure can be identified from work that studies analogies or linear subspaces encoding bias.  These have been shown in multiple languages.
>
>
> ***C5: Unsupported claims (Part 2)***
>
> 2.  “effectively removes much of the clumping of words based on properties other than the pairwise similarity which encodes meaning.” - this term is not introduced or explained as a consequence or issue within CLWE prior art. It is unclear what this means, or how it is required to be mitigated for better CLWE.
>
> ***RESPONSE:***
>
> Perhaps clustering is a more informative word than clumping.  We also hope the added Figure 1 helps clarify what is meant.
>
>
> ***C5: Unsupported claims (Part 3)***
>
> 3.  “This buttresses the claim made by Xu & Koehn (2021) that improving the degree of isomorphism” - it is unclear where you demonstrate that you improve isomorphism with your method. If your iterative method subsumes methods already maintaining/improving isomorphism then it would be beneficial to demonstrate that such properties are maintained. If this is intended to be obvious, please state so.
>
>  ***RESPONSE:***
>
> We feel that improved isomorphism by our methods is demonstrated in two general ways.  First, after I-C+SN+L normalization, then the embeddings are all centered, have unit-length words, and have similar spectral properties.  Second, we are able to achieve rigid alignments which are more useful for cross-lingual tasks, our ultimate goal.  If there was not some sort of structural isomorphism in these embeddings (that our normalization enhances) then these cross-lingual (and other cross-embedding approaches) would probably not work at all.
>
>
>
> ***C5: Unsupported claims (Part 4)***
>
> 4.  “Our approach generalizes previous approaches” -- this is not proven (see the previous comment), and little care is taken to show that SN does not interfere with complementary components like Length Normalization.
>
>     ***RESPONSE:***
>
> This is referring two the I-C+SN+L method which also ensures center normalization and length normalization.  Moreover, as discussed in response to another reviewer, by adjusting the parameters of spectral normalization, it can achieve something very similar to another approach: PCA removal.
>
>
> ***C5: Unsupported claims (Part 4a)***
>
> -   Is Isomorphism maintained and can you verify that your method subsumes prior methods with absolute certainty?
>
>
> ***RESPONSE:***
>
> Perhaps you are referring to "self-isomorphism," asking if the normalization retains the key properties of the original embedding.
> In that sense, none of the preprocessing methods attain absolute isomorphism.  As discussed in the paper, length normalization distorts Euclidean distance and center normalization distorts cosine distance.  Similarly, spectral normalization distorts both.  However, following previous work, we have observed that it does not significantly degrade performance on inter-embedding tasks, yet improves alignment and cross-embedding tasks.
>
> Moreover, Table 2 shows that the performance actually improves on monolingual word similarity tasks.  We are not arguing this is a significant change, but it is not decreasing the ability to detect similarities.  These are fairly standard stand-alone tasks to evaluate the effectiveness of word embeddings. The fact that it is not hurting this score indicates that some *relevant* aspects of self-isomorphism are retained.
>
>
>
> ***C5: Unsupported claims (Part 4b)***
>
> -   What is meant by clumping?
>
> ***RESPONSE:***
>
>   Clumping here means clustering.  Hopefully, the new Figure 1 helps clarify this.
>
>
>
> ***C5: Unsupported claims (Part 4c)***
>
> -   What are the motivating failings of prior work that has led to the SN method?
>
> ***RESPONSE:***
>
> As discussed in the paper, other works either did not try to address the spectral disparities between languages before align or like PCA removal did so in a blunt way.
>
> ***C5: Unsupported claims (Part 4d)***
>
> -   What is the intended intuition and understanding of “regularizing an embedding space”?
>
>  ***RESPONSE:***
>
> We attempted to address this via the above comments and the new Figure 1.

---

> ### Author Response · Authors · 2021-11-19
> **Response to Reviewer zhG1**
>
> ***C3 & C4:***
>
>
> 1.  Care is not taken to explicitly link together the spectral statistics to additional methods or to explain either what spectral metrics are informing the reader of in 2.2. It is somewhat of a foregone conclusion that spectrum-based metrics will improve with a spectrum-based method. While this section relies on the insight of Dubossarsky et al. (2020), there is little effort in interrogating the prior work and motivating why prior insight is directly applicable here as-is. At present, this must be taken at face value with little motivation for the succeeding sections. This caveat is somewhat improved in Appendix B, but I consider this insufficient without a clear hypothesis tying the motivation of this approach together.
>
>
> 2.  Overall, there is a lack of research questions in the work which leads to a difficulty in scoping the contribution. SN is clearly useful -- but why? We observe empirical benefits to several tasks, but the work does not source these benefits to specific components or processes within the method. This could be improved with intrinsic analysis (and some error analysis) identifying the effect of SN. This is hinted at in Section 4 where it is stated that “We hypothesize this is because it is somewhat uniformly stretching words along the major modes of variation” however this could be expanded to aid in understanding of the mechanics of the method.
>
> ***RESPONSE:***
>
> We attempted to respond to these concerns above and in varous other responses.

---

> ### Author Response · Authors · 2021-11-19
> **Response to Reviewer zhG1**
>
> ***C2:***
> The work does not make a self-contained case for what spectral normalization is a solution towards. The problem is poorly defined, which leads to a gap in understanding the motivating strategy of the work. For example, in Section 3 the work states “As a result, if the spectral properties are extreme, it can help regularize them.” -- but it is not clear what it means to (i) regularize such properties or (ii) what this intends to do with respect to the embedding spaces. This could be improved if the problem hypothesis was clearer e.g. answering questions such as:
>
> -   Are spectral properties of monolingual embedding spaces malformed?
>
> -   Can this be demonstrably proved in this work so the problem to be solved is clear?
>
> -   Are there confounding examples within embedding spaces to motivate this (e.g. unexpected versions of the common king-queen=man-woman analogies based on low-frequency words)?
>
> -   What does spectral normalisation hope to intrinsically accomplish such that the findings can be better interpreted?
>
> For example, in Section 3 the work states “As a result, if the spectral properties are extreme, it can help regularize them.” – but it is not clear what it means to (i) regularize such properties or (ii) what this intends to do with respect to the embedding spaces.
>
>
> ***RESPONSE:***
>
> We are not claiming that embeddings are malformed, the same way not being length-normalized or not centered does not make them malformed.  It just makes different embeddings (from different sources or learned with different techniques, but capturing the same phenomenon, e.g., human language) hard to align.
>
> We review in a comment to *Reviewer ve8X* some properties from (https://arxiv.org/pdf/2001.11136.pdf) that captures how when languages have similar spectral properties, and when they happen to align better.
>
>
> Consider the following fictional thought experiment which might shed some light as why Spectral Normalization may be useful.
> Some embeddings have the ability to capture analogies such as man:woman::king:? as queen.  This works by adding the vector woman-man to king, and finding the nearest neighbor.  This somewhat implies linear subspaces, one capturing gender and one capturing royalty.  Now imagine two languages that have these two subspaces encoded, but one has many more gendered words, and that direction is more clearly captured; hence the words along this direction are more stretched out.  Another has less gender encoded, but perhaps a monarchy and a lot of monarchy-related terms in the corpus, so the royalty related direction is comparatively more well captured, and hence stretched out.  The different stretching of these directions may interfere with alignment (as measured by Dubossarsky et al. (2020)).  For instance, these elongated directions (corresponding with larger singular values) may be pulled to align with each other, even though they represent different concepts.
>
> Another attempt to illustrate this more concretely is shown in the newly added Figure 1 in the paper.  This shows how after center and length normalization, two languages can have similar language patterns (we can only draw in 2d, so we show this with ordering around the circle), but different clustering of those orderings.  This leads to an inability to find a very good alignment.  With Spectral Normalization, such clumping cannot occur, or at least not in such an imbalanced way, and in this examples forces the words to be spread out.  After this step, the languages can then be well aligned with each other.
>
> Thus Spectral Normalization can provide another form of embedding geometry regularization.  Our hypothesis is that these occur due to language-specific property, but not meaning or structure specific reasons, and that Spectral Normalization will adjust the embeddings to remove this differences.  We cannot directly verify or quantify these notions, but we can evaluate their performance under alignment.  And indeed, the alignment and alignment-requiring tasks improve performance with Spectral Normalization.

---

> ### Author Response · Authors · 2021-11-19
> **Response to Reviewer zhG1**
>
> ***C1:***
> This method shows merit for the reasons given above, but I can describe my primary concern as this — you have proposed solution X for some method A, claiming that A is now improved due to a reduction in effect Z. But you do not demonstrably support that solution X influences effect Z or highlight if effect Z is a critical failure in A. Therefore, while empirical results are strong, it remains unclear what solution X does to improve A. Even if the influence of X on A is not well understood -- this should be delineated further to support that the empirical claims require further introspection.
>
> ***RESPONSE:***
>
> We are a bit unsure of the intended meaning for X, A, and Z.  Let us attempt to explain the core of the logic and motivation.
>
> The ultimate goal is better alignments of embeddings; we state this vague to include the many non-language scenarios where we do not have as good data.  We can (and do) measure this empirically using MRR of translation pairs (or similar measures) and in two downstream tasks.  Improving the performance of these tasks is our best indication of a better methodology.
>
> We propose a new method (Spectral Normalization), based on evidence from recent papers that the spectral properties of embeddings correspond with how well they can be aligned (using evidence like MRR scores on cross-embedding similarity tasks to evaluate this).  This work does not say adjusting these spectral properties will help, this is our new hypothesis, and the proxy goal of our method.  And it motivated us to try the proposed approach.
>
> We verify that applying Spectral Normalization does improve the performance on the evaluation tasks (including MRR of translation pairs).
>
> One may still wonder *why* exactly spectral normalization works from a structural perspective.  We are a bit less clear on this, and it is harder to quantify.  We attempt to provide some insight in the response to the next concern.

---

> ### Author Response · Authors · 2021-11-19
> **General Comments**
>
> We thank the reviewers for their insightful comments.  Sorry for the slow reply, we ran a significant number of new experiments, based on your comments, which took time to complete.
> Towards addressing the comments we have updated the paper in three significant ways.  (As a result, some other parts were trimmed.)
>
>
> 1.  We inserted an example (Figure 1, which appears in Section 3) which attempts to illustrate some of the intuition behind how Spectral Normalization can help with alignment, in ways the verbal description in the paper seems to have failed to adequately convey.  It shows a toy example in 2d with corresponding words in English and Japanese.  After centering and length normalization, all words live on the unit circle.  Despite the same ordering of words, the ones in Japanese have ones clustered at the top and bottom of the unit circle, while in English they are clustered on the left and right.  This prevents any great orthogonal alignment.  However, with Spectral Normalization also applied, the words cannot be clustered in the same way, and are made more spread out, while still on the unit circle.  This allows for a much better alignment.
> While the effects of Spectral Normalization in practice are quite likely much more subtle when considered in higher dimensions and on the full data sets (although the spectral properties of English and Japanese are extremely different, so perhaps not), we hope this provides the missing intuition about how Spectral Normalization could be helping.
>
>
> 2.  We added several experiments using the I-C+L normalization as a baseline, this included in the contextual embedding comparison (Table 5), the Cross-lingual document classification task (Table 6), the cross-lingual natural language inference task (Table 7), Alignment of Single-cell Multi-omics Data (Table 17 in Appendix G), and BLI performance comparison on all the 28 language pairs in the GTrans dictionary (Tables 20, 21 in Appendix K, and their summary results in Table 15 in Appendix E).  Similar to the results on the held-out language pairs (Table 3), the I-C+SN+L normalization mostly out-performs I-C+L normalization, but not as much the no normalization baseline.  There are a few exceptions, notable RCSLS which learns an affine transform in addition to the rigid transformation; although in most cases I-C+SN+L still performs the best.  In some cases, the non-rigid transform learned by RCSLS may be inappropriate, or not as effective as in the XNLI task.  Overall, these extensive additional experiments with the additional I-C+L baseline further confirm the helpfulness of the Spectral Normalization.
>
>
> 3.  We attempted to quantify the significance of the improvement gained by Spectral Normalization.  While the numerical improvement of including this approach in addition to centering and length normalization is often small, it is quite consistent.  Measuring a statistical significance on any one task is difficult also since the techniques and experiments are deterministic.  However, we did show significance in the consistency of the improvement using a Binomial test.  This is shown in Appendix E.  For instance, in Table 15, using PROC-B alignment, I-C+SN+L normalization improves over I-C+L normalization on 25 of 28 languages.  Under the Binomial test, this has a p-value of $0.00001$, and is significant at reasonable significance levels (e.g., 0.01).

---

> > ### Comment · Reviewer_zhG1 · 2021-11-29
> > **Response to authors (general and review-specific)**
> >
> > I thank the authors for their efforts in their response. Responding to 5 reviews in a timely manner is no small feat and the submission remains extensive in its exploration.
> >
> > Figure 1 is useful in illustrating your expectations and intentions with regards to SN. Improvements on significance testing and baseline comparisons both aid in the experimental rigor of your work.
> >
> > I remain concerned about the motivation and hypotheses in the work. In your response, you state a candidate hypothesis and list reasons why your hypothesis cannot be tested. Therefore, I would argue that this is not a true hypothesis and the experimental framing of this work should be improved to analyze a legitimate hypothesis that can be proven or disproven. Other reviewers share concerns surrounding the framing of this work and your responses have aided in the overall understanding of the contribution. However, such explanations should really be inside the work to strengthen the contribution as a scientific proposal and not a technique that just so happens to work well. I am not inclined to modify my rating as I believe this work could benefit from further revisions to be a strong candidate at a later venue.

---

### Official Review · Reviewer_GCj8 · 2021-11-03

**Correctness:** 3
**Technical Novelty And Significance:** 2
**Empirical Novelty And Significance:** 2
**Recommendation:** 3
**Confidence:** 4

**Main Review:**

SpecNorm seems like a very reasonable idea, but it takes a long time to finally present it on page 4 (out of 9). I also don’t think that pseudocode for SpecNorm is appropriate; it’s hard to understand, but the verbal explanation is far clearer.

Are you able to include some intuition for why capping the singular values is helpful? I gather that some of this is given by Dubossarsky et al. (2020), but to make this paper self-contained, it would be helpful to provide some intuition here as well.

I-C+SN+L is a more complicated method. How did you decide on these three normalizations, and their order? Why does iterating help? It would be nice if the method could be better motivated.

The five results tables compare four different sets of systems. It's especially confusing that sometimes SN means SpecNorm and sometimes it means I-C+SN+L. Besides being difficult to follow, it creates a feeling that the results are selectively reported. If possible, please include the same set of normalizations for all experiments.

A much more minor suggestion is that Table 4 should be transposed, so that the compared normalizations are the rows, as in all the other tables.

**Summary Of The Paper:**

This paper describes three new methods for normalizing word embeddings before cross-lingual alignment. The first method, SpecNorm, caps the singular values of the word embeddings to be twice the average (original) singular values. The second method does mean centering, SpecNorm, and L2 normalization. The third repeats this for a fixed number of iterations (5). Experiments on a wide variety of tasks generally show improvements.


**Summary Of The Review:**

The core SpecNorm method seems reasonable, but the iterative centering + SpecNorm + length normalization seems more ad hoc. Although the experiments are thorough and show improvements, I would like to see more principled motivation for method and/or deeper insight into why it works.

---

> ### Author Response · Authors · 2021-11-19
> **Response to  Reviewer GCj8**
>
> ***C3:***
> I-C+SN+L is a more complicated method. How did you decide on these three normalizations, and their order? Why does iterating help? It would be nice if the method could be better motivated.
>
> ***RESPONSE:***
>
> Unit-length and zero-mean normalization have been previously shown effective (https://aclanthology.org/D16-1250.pdf, https://aclanthology.org/P19-1307.pdf).
> Also, PCA removal has been a heuristic independently considered.  Recently Dubossarsky et al. (2020) showed that spectral differences in languages had strong correlation with how well they could be aligned.  So it seemed natural to try to combine all of these approaches.
>
> The ordering of them is what makes the most sense.
>  - Spectral normalization makes the most sense on data that has been centered already (like how PCA first centers data before apply low-rank approximation)
>  - Spectral normalization does not change the centering property of the data.
>  - Unit-length normalization changes the mean of the data, so the data is no longer centered after it has been applied.  It also affects the spectral properties, as Spectral normalization affects the unit length normalization.
>  - Centering removes the unit-length property, so toward achieving all goals, it should come after unit-length normalization.
>
>  Hence, overall, to accommodate all of these properties, the logical order is C -> S -> L -> C -> ...
>
>
>
>
> ***C4:***
> The five results tables compare four different sets of systems. It's especially confusing that sometimes SN means SpecNorm and sometimes it means I-C+SN+L. Besides being difficult to follow, it creates a feeling that the results are selectively reported. If possible, please include the same set of normalizations for all experiments.
>
> ***RESPONSE:***
>
> We have attempted to update the language to make the distinction between normalization approaches more clear.
>
> It takes a while to run these experiments, so we are unable to include all variants, but as discussed above, we have included results for the main baseline, I-C+L normalization, to a large number of experiments for this rebuttal.

---

> ### Author Response · Authors · 2021-11-19
> **Response to Reviewer GCj8**
>
>   ***C1:***
> SpecNorm seems like a very reasonable idea, but it takes a long time to finally present it on page 4 (out of 9). I also don’t think that pseudocode for SpecNorm is appropriate; it’s hard to understand, but the verbal explanation is far clearer.
>
> ***RESPONSE:***
>
> There have been dozens of papers in the last few years on this topic, and we felt it necessary to put the work in context.  This took a few pages.
>
> Some readers prefer textual descriptions, and some more are aided by pseudocode.  We believe this algorithm is the main contribution, and that it deserves the highlight of the short pseudocode.  So we have included both.
>
> The main Spectral normalization algorithm is also contained in the Appendix in included Spectral Normalization python code (SVDConX.py) and the Iterative Spectral Normalization algorithm is in (Iter_SVDConX.py).
>
>
> ***C2:***
> Are you able to include some intuition for why capping the singular values is helpful? I gather that some of this is given by Dubossarsky et al. (2020), but to make this paper self-contained, it would be helpful to provide some intuition here as well.
>
> ***RESPONSE:***
>
> There is some discussion addressing this in Dubossarsky et al. (2020), but the lead up to the new algorithmic contribution is already more than 3 pages.  As mentioned in the general comments, we added a new Figure 1 attempting to illustrate the sort of effects that Spectral Normalization may be having that could improve the alignment.
>
> Here is more detail on the relations been spectral properties of embeddings, and their construction and generating data:
>
> Some of the reasons that account for the structural differences or typological differences between the two monolingual embedding spaces include
>  - differing word frequency,
>  - corpus size on which embedding induction algorithms are trained,
>  - training duration of the embedding induction algorithm,
>  - topical skewness of monolingual corpus used to train the monolingual embeddings.
> These differences seem to correlate with the ability of Cross-lingual word embedding methods to learn a good transfer function from one monolingual embedding space to the other robust to noise.
> The smaller the (effective) condition number of two embedding spaces, the more robust it is to learn a transfer function mapping one embedding space to another (https://arxiv.org/pdf/2001.11136.pdf). Smaller singular values are associated with noise and the least amount of semantic information. So the goal of the Spectral Normalization is to determine the effective dimensionality or rank of the monolingual embedding space and then set all the singular values below the effective dimensionality, or rank to 1 since these are the noisy singular values and then scale the singular values above the effective dimensionality, or rank with $\frac{1}{\sigma_{i}}\times\left(\beta\times\eta\right)$ where $\eta$ is the square root of the average squared singular values, $\beta\geq1$ and ${\sigma_{i}}$ is the $i-$th singular value and which is the. $\eta$ and $\beta$ are used to determine the smaller singular values associated with noise or with the least important information that has to be removed.  When $\beta=0$, much emphasis, and weight are given to the noisy part of the data, this corresponds to the tail of the singular component, which has smaller principal values associated with them. Our solution to this is to make $\beta\geq1$, where we only regularize the word embeddings with the larger principal values. $\beta$ is introduced as a support to help get rid of much of the noise. It is chosen through cross-validation on a selected language pair for the BLI task. Removing the smaller singular values associated with noise improves the condition number of the monolingual embedding. Which in turn helps to learn a transfer function that is robust to noise. In other to get rid of the smaller singular values associated with noise we transform the diagonal matrix, $\Sigma$ to $\Sigma^{\prime-1}$ this way, for each singular value $\sigma_{i}$, if  $\Sigma_{ii}>\left(\beta\times\eta\right)$ we substitute $\sigma_{i}$ with $\frac{1}{\sigma_{i}}\times\left(\beta\times\eta\right)$ else we make $\sigma_{i} = 1$. The affine scaling $\frac{1}{\sigma_{i}}\times\left(\beta\times\eta\right)$ reduces the percentage of variance explained by the highest eigenvalue. This prevents the monolingual word embedding from being stretched out along the direction of the eigenvector with the highest eigenvalue so that they are not clustered around the eigenvector with the highest eigenvalue.

---

> ### Author Response · Authors · 2021-11-19
> **General Comments**
>
>
>
> We thank the reviewers for their insightful comments.  Sorry for the slow reply, we ran a significant number of new experiments, based on your comments, which took time to complete.
> Towards addressing the comments we have updated the paper in three significant ways.  (As a result, some other parts were trimmed.)
>
>
> 1.  We inserted an example (Figure 1, appears in Section 3) which attempts to illustrate some of the intuition behind how Spectral Normalization can help with alignment, in ways the verbal description in the paper seems to have failed to adequately convey.  It shows a toy example in 2d with corresponding words in English and Japanese.  After centering and length normalization, all words live on the unit circle.  Despite the same ordering of words, the ones in Japanese have ones clustered at the top and bottom of the unit circle, while in English they are clustered on the left and right.  This prevents any great orthogonal alignment.  However, with Spectral Normalization also applied, the words cannot be clustered in the same way, and are made more spread out, while still on the unit circle.  This allows for a much better alignment.
> While the effects of Spectral Normalization in practice are quite likely much more subtle when considered in higher dimensions and on the full data sets (although the spectral properties of English and Japanese are extremely different, so perhaps not), we hope this provides the missing intuition about how Spectral Normalization could be helping.
>
>
> 2.  We added several experiments using the I-C+L normalization as a baseline, this included in the contextual embedding comparison (Table 5), the Cross-lingual document classification task (Table 6), the cross-lingual natural language inference task (Table 7), Alignment of Single-cell Multi-omics Data (Table 17 in Appendix G), and BLI performance comparison on all the 28 language pairs in the GTrans dictionary (Tables 20, 21 in Appendix K, and their summary results in Table 15 in Appendix E).  Similar to the results on the held-out language pairs (Table 3), the I-C+SN+L normalization mostly out-performs I-C+L normalization, but not as much the no normalization baseline.  There are a few exceptions, notable RCSLS which learns an affine transform in addition to the rigid transformation; although in most cases I-C+SN+L still performs the best.  In some cases, the non-rigid transform learned by RCSLS may be inappropriate, or not as effective like in the XNLI task.  Overall, these extensive additional experiments with the additional I-C+L baseline further confirm the helpfulness of the Spectral Normalization.
>
>
> 3.  We attempted to quantify the significance of the improvement gained by Spectral Normalization.  While the numerical improvement of including this approach in addition to centering and length normalization is often small, it is quite consistent.  Measuring a statistical significance on any one task is difficult also since the techniques and experiments are deterministic.  However, we did show significance in the consistency of the improvement using a Binomial test.  This is shown in Appendix E.  For instance, in Table 15, using PROC-B alignment, I-C+SN+L normalization improves over I-C+L normalization on 25 of 28 languages.  Under the Binomial test, this has a p-value of $0.00001$, and is significant at  reasonable significance levels (e.g., 0.01).

---

### Official Review · Reviewer_ve8X · 2021-11-03

**Correctness:** 4
**Technical Novelty And Significance:** 3
**Empirical Novelty And Significance:** 3
**Recommendation:** 8
**Confidence:** 4

**Main Review:**

Strength:
+ The goal is clearly presented and the method is clean (and clear)
+ The paper is well-organized and clearly written
+ Experimental results show consistent improvement

Weaknesses:
- Limited to preprocessing for a specific cross-lingual embedding setup
- No uncertainty/confidence/error bars on experimental results, or significance testing [addressed]
- Some experiments may use a weak baseline [addressed]

The proposed normalization improves performance quite consistently overall and on average, even if it sometimes slightly degrades performance on few specific settings. This is an excellent sign that the spectral normalization does something reasonable and tends to help.
However in many experimeents, the compared baseline is simply no normalization, which is really not a very strong opposition. It is not surprising that some normalization helps vs. no normalization at all. In fact BLI results like Table 3 suggest that there is a much larger difference between any normalization and no normalization, than there is between the various normalizations tested. This raises the question of how the proposed SN compares to *other normalizations*, especially on downstream tasks.
In addition, the conclusion and Appendix E suggest that training joint cross-lingual embeddings compares favourably with the normalization+rigid transformation approach, yet only the latter is considered in the main paper.

In Figure 1: In addition to the desirable impact on effective condition numbers and single value gap on all languages, the iterative spectral normalization seems to have a massive impact on the effective rank for Japanese, taking it remarkably close to the ER for other languages (wherease the raw er was much lower according to the log scale). Any idea where this large effect comes from or whether this is due to the particular language or dataset?

Re. stability of language choice: experiments are run on different languages at different points in the paper. For example Hindi and Japanese are used in Figure 1, but not later. The choice of the four particular language pairs used in the cross-lingual natural language inference experiments is justified, however it is unclear why some BLI results use AR, DE and NL (Tab. 5), others include IT (Tab. 6).

Questions:

"significantly improves over the baseline" (p.7): What kind of significance testing was run?

"Normalizing contextual type-level embedding" (p.8): It is not clear what was done here. I understand some details are available in Xu&Koehn(2021), but this paper should arguably provide enough information to be self-contained and understandable.

In what way does the proposed spectral normalization "generalize previous approaches" (p.9)? Are there specific choices of settings for which it becomes equivalent to centering, length normalization or PCA removal? This is not obvious, especially since some of these have guarantees (eg on preserving angle or relative distances) that SN does not have.

Typos:
* P2,l14: "Euclidean of cosine" - presumably "or"
* P3,l2: what is "wlog"
* P6,l3: publically
* P7,l14: "with C+L" - presumably "without" is meant
* Tab3: $X_L_1$ should be $X_L_2$?
* Tab4,6,7,13,18,19: 5K, 3K, 1K should be 5k, 3k, 1k (lowercase)

[Thanks for the extensive reply to this and other reviewer's comments -- much appreciated]

**Summary Of The Paper:**

The paper proposes a new spectral normalization technique that improves the cross-lingual mapping of monolingual embeddings by rigid, orthogonal transformations. This is demonstrated by consistent gains on bilingual lexical induction and two other downstreeam cross-lingual tasks.


**Summary Of The Review:**

A simple, clean preprocessing method to help map word embeddings across languages. The article is clear and well organized, and extensive experiments produce consistent gains in a number of cross-lingual tasks. The experimental setup raises a few questions regarding the significance of these improvements.

---

> ### Author Response · Authors · 2021-11-19
> **Response to Reviewer ve8X**
>
> ***C6:***
> Re. stability of language choice: experiments are run on different languages at different points in the paper. For example, Hindi and Japanese are used in Figure 1, but not later. The choice of the four particular language pairs used in the cross-lingual natural language inference experiments is justified, however, it is unclear why some BLI results use AR, DE, and NL (Tab. 5), others include IT (Tab. 6).
>
> ***RESPONSE:***
>
> **Translation Dictionary: In this paper we used two type of translation dictionaries:**
>
> -  **MUSE translation dictionary** (https://arxiv.org/pdf/1710.04087.pdf) has 88 translation pairs that involve English as either the source or target language.
> -  **GTrans translation dictionary**  (https://arxiv.org/pdf/1902.00508.pdf) which has 28 language pairs spanning 8 different languages. GTrans translation dictionary is more diverse since 75% of the translation pairs do not involve English.
> This explains why for some of the tables, we do not see some language pairs.
>
> **Explanation of the experiments set up in the paper:**
>
> 1. **Figure 1:** Our goal was to show the impact of iterative spectral normalization in improving the degree or level of isomorphism through several Approximate Isomorphism measures like Effective Rank, Singular value gap, Effective condition number, and Joint Effective condition number. We picked English and German from the **set of similar European languages** and Japanese and Hindi from the **set distant languages**.
> 2. **Hyperparameter Tuning, Table 1&2:** To avoid overfitting, we performed cross-validation on a held-out set of **5 languages** English (EN), Hindi (HI), Russian (RU), Chinese (ZH), Japanese (JA), Turkish (TR). **Ten (10) Language pairs** of the form EN-X and X-EN were used for hyperparameter tuning for iterative spectral normalization algorithm. We used the publicly available ***MUSE translation dictionary*** for hyperparameter tuning.
> 3. **Comparison of Iterative Spectral Normalization to other normalization algorithms from previous works on the BLI task, Table 3:** We also **MUSE translation dictionary**. Specifically, we evaluated 1**8 language pairs**, i.e., English (EN) from/to Bulgarian (BG), Catalan (CA), Czech (CS), German (DE), Spanish (ES), Korean (KO), Thai (TH) and Chinese (ZH) separate from the language we used for the hyperparameter tuning (details in Appendix I).
> 4. **Comprehensive study of Performance of Iterative normalization on the BLI task, Table 4 and 13:** Here, we use the **Google Translate (GTrans) dictionary**. It includes 28 language pairs spanning 8 different languages: Croatian (HR), English (EN), Finnish (FI), French(FR), German (DE), Italian (IT), Russian (RU), and Turkish (TR). We trained on 1k, 3k, and 5k source words and evaluated on 2k source test queries (details in Appendix D and K).
> 5. **Comparison of Iterative Spectral Normalization to Iterative normalization algorithm (the best) from previous works on the BLI task:** Similarly, we use the **Google Translate (GTrans) dictionary**. It includes 28 language pairs spanning 8 different languages: Croatian (HR), English (EN), Finnish (FI), French(FR), German (DE), Italian (IT), Russian (RU), and Turkish (TR). We trained on 5k source words and evaluated 2k source test queries (details in Appendix J).
> 6. **BLI performance (MAP) on aligning Cross-lingual Contextual Embedding, Table 5** We compared the impact of Iterative Normalisation and Spectral Normalisation on the BLI performance on aligning Cross-lingual Contextual Embedding. We used the baseline result from (https://arxiv.org/pdf/2107.09186.pdf), which was evaluated on English (EN), German (DE), Arabic (AR), and Dutch (NL).
> 7. **Cross-lingual Document Classification (CLDC), Table 6:** We evaluated the resulting CLWE on a cross-lingual document classification (CLDC) task. The TED CLDC corpus was used for training and evaluation. It contains 15 topics and 12 language pairs. The intersection between TED CLDC corpus languages and our BLI languages **(GTrans)** resulted in the following five CLDC evaluation pairs: EN–DE, EN–FR, EN–IT, EN–RU, and EN–TR.
> 8. **Cross-lingual Natural Language Inference (XNLI), Table 7:** We used a multi-lingual XNLI corpus which includes 5 of the 8 languages used in the BLI task **(GTrans)**: EN, DE, FR, RU, and TR.

---

> > ### Comment · Reviewer_ve8X · 2021-11-30
> > **Thanks for the response**
> >
> > It is clear that the authors put a lot of work in replying to the reviews, providing a large amount of additional experimental results and many clarifications. The revised paper also provides a better intuition about the method and significance testing, which support the conclusion that the method provides a consistent boost in performance. I am still positive about this paper and updated my review accordingly.
> > Thanks for the extensive reply!

---

> ### Author Response · Authors · 2021-11-19
> **Response to Reviewer ve8X**
>
> ***C4:***
> "significantly improves over the baseline" (p.7): What kind of significance testing was run?
>
> ***RESPONSE:***
> As discussed in the general comment, we had added significance scores using a Binomial test.
> The updated paper attempts to use the word "significant" more judiciously.
>
>
>
> ***C4:***
> "Normalizing contextual type-level embedding" (p.8): It is not clear what was done here. I understand some details are available in Xu&Koehn(2021), but this paper should arguably provide enough information to be self-contained and understandable.
>
> ***RESPONSE:***
> We have included a new Appendix Section D that explains this in more detail.
>
> Fast Align (https://aclanthology.org/N13-1073/) is applied to parallel corpora to derive silver aligned token pairs. The contextual embeddings are obtained by feeding the tokenized parallel corpus into pretrained BERTs of the source and target language. Since there will be multiple occurrences of type-level words and each type-level word will possess a contextual word embedding, for each type-level word, its vector representation is derived from the mean vector of all the type-level words from the monolingual corpus fed into the pre-trained language model. This is done for both source and target language. The resulting type-level contextual embedding from the source and target language are aligned by solving the Procrustes problem.
>
>
> ***C5:***
> In what way does the proposed spectral normalization "generalize previous approaches" (p.9)? Are there specific choices of settings for which it becomes equivalent to centering, length normalization, or PCA removal? This is not obvious, especially since some of these have guarantees (eg on preserving angle or relative distances) that SN does not have.
>
> ***RESPONSE:***
>
> From Algorithm 2, when we set beta to a very large number, the else statement within the for loop is triggered all the time. Thereby setting all the diagonal elements to 1. This means no form of affine scaling is taking place. So lines 2 and 4 will be the only preprocessing steps that will be carried out. And this is the same as the original Iterative normalization algorithm.
>
> Similarly, when we turn off lines 2 and 4 and only perform 3 we achieve roughly the same effect of PCA removal by reducing the top principal component or top singular vector.

---

> ### Author Response · Authors · 2021-11-19
> **Response to Reviewer ve8X**
>
> ***Concern 1 (C1):***
> However, in many experiments, the compared baseline is simply no normalization, which is really not a very strong opposition.
>
> In fact, BLI results like Table 3 suggest that there is a much larger difference between any normalization and no normalization than there is between the various normalizations tested. This raises the question of how the proposed SN compares to other normalizations, especially on downstream tasks.
>
> ***RESPONSE:***
>
> Our experiments demonstrated that I-C+SN+L was the best normalization approach, and then showed the performance of this normalization against not normalizing to show how much improvement could be realized.  But this is a reasonable comparison to request, and we ran a significant number of new experiments using the I-C+L normalization baseline.  See the general discussion points.
>
> As with the BLI task, the I-C+SN+L normalization typically improved on the I-C+L normalization, but not as much as the no normalization baseline.  A perhaps notable exception is with the non-rigid alignment RCSLS.  While I-C+SN+L often does see improvement on tasks over I-C+L, it does not always as consistently with RCSLS.  In particular, in the CLDC task, I-C+L does better on 3/5 language pairs, and better on average.  However, in other downstream tasks like XNLI and genomic data, I-C+SN+L always does better across all alignment approaches.
>
>
> ***C2:***
> In addition, the conclusion and Appendix E suggest that training joint cross-lingual embeddings compares favorably with the normalization+rigid transformation approach, yet only the latter is considered in the main paper.
>
> ***RESPONSE:***
>
> In short, the main advantage of our proposed normalization+rigid transformation is that it is a simple post-processing that does not require the expensive step of retraining of the entire embeddings.  Moreover, due to the details of how joint-training of embeddings works, it may not always be an option.
> Hence, our primary focus is on normalization for alignment of fixed embeddings in this paper.
>
>
>
>
>
> ***C3:***
> In Figure 1: In addition to the desired impact on effective condition numbers and single value gap on all languages, the iterative spectral normalization seems to have a massive impact on the effective rank for Japanese, making it remarkably close to the ER for other languages (whereas the raw er was much lower according to the log scale). Any idea where this large effect comes from or whether this is due to the particular language or dataset?
>
> ***RESPONSE:***
>
> Several reasons have been observed to account for the structural differences or typological differences between two monolingual embedding spaces. Amongst these reasons are:
>  -  Differing word frequency
>  - Corpus size on which embedding induction algorithms are trained on
>  -  Training duration of the embedding induction algorithm
>  -  Topical skewness of monolingual corpus used to train the monolingual embeddings
>
> While it may be hard to find out which factor(s) account for specific structural differences between two monolingual embedding spaces, recent work has shown that the spectral statistics largely correlate with the map-ability of the two spaces (https://arxiv.org/pdf/2001.11136.pdf). Embedding space with similar singular values tends to align better than space with differing singular values. For instance looking at the percentage of variance explained by the highest eigenvalue for these set of languages (https://arxiv.org/pdf/2011.14874.pdf):
>  -  English, EN: 6.9%
>  -  Spanish, ES: 5.6%
>  -  French, FR: 4.3%
>  -  Italian, IT:  4.5%
>  -  German, DE: 4.4%
>  -  Chinese, ZH: 49.2%
>  -  Japanese, JA: 96.71%
>  -  Vietnamese, VI: 8.6%
>  -  Thai, TH: 8.6%
>
> we see that for similar languages like English, Spanish, French, Italian and German, the percentage of variance explained by the highest eigenvalue is small compared to distant languages like Chinese, Japanese, Vietnamese, and Thai. Japanese, JA, in particular, has about 96% of the variance explained by the highest eigenvalue. This means that most Japanese word embedding will be stretched out along the direction of the eigenvector with the highest eigenvalue and hence clustered around the eigenvector with the highest eigenvalue when centered and length normalized, making learning a good and robust transfer function very difficult. This explains why EN-JA yields a P@1 of 2%.
>
> This is where our new preprocessing algorithm comes to bear. The goal of iterative spectral normalization is to make the direction of the monolingual embeddings in the embedding space uniformly distributed, aside from ensuring that the monolingual embeddings have a unit length and zero mean which is very useful under orthogonal mapping.
> Spectral normalization reduces the percentage of variance explained by the highest eigenvalues (say for Japanese).  This change also corresponds with reducing the effective rank, which is an indicator (but perhaps not direct cause) of this possible issue.

---

> ### Author Response · Authors · 2021-11-19
> **General Comments**
>
> We thank the reviewers for their insightful comments.  Sorry for the slow reply, we ran a significant number of new experiments, based on your comments, which took time to complete.
> Towards addressing the comments we have updated the paper in three significant ways.  (As a result, some other parts were trimmed.)
>
>
> 1.  We inserted an example (Figure 1, which appears in Section 3) which attempts to illustrate some of the intuition behind how Spectral Normalization can help with alignment, in ways the verbal description in the paper seems to have failed to adequately convey.  It shows a toy example in 2d with corresponding words in English and Japanese.  After centering and length normalization, all words live on the unit circle.  Despite the same ordering of words, the ones in Japanese have ones clustered at the top and bottom of the unit circle, while in English they are clustered on the left and right.  This prevents any great orthogonal alignment.  However, with Spectral Normalization also applied, the words cannot be clustered in the same way, and are made more spread out, while still on the unit circle.  This allows for a much better alignment.
> While the effects of Spectral Normalization in practice are quite likely much more subtle when considered in higher dimensions and on the full data sets (although the spectral properties of English and Japanese are extremely different, so perhaps not), we hope this provides the missing intuition about how Spectral Normalization could be helping.
>
>
> 2.  We added several experiments using the I-C+L normalization as a baseline, this included in the contextual embedding comparison (Table 5), the Cross-lingual document classification task (Table 6), the cross-lingual natural language inference task (Table 7), Alignment of Single-cell Multi-omics Data (Table 17 in Appendix G), and BLI performance comparison on all the 28 language pairs in the GTrans dictionary (Tables 20, 21 in Appendix K, and their summary results in Table 15 in Appendix E).  Similar to the results on the held-out language pairs (Table 3), the I-C+SN+L normalization mostly out-performs I-C+L normalization, but not as much the no normalization baseline.  There are a few exceptions, notable RCSLS which learns an affine transform in addition to the rigid transformation; although in most cases I-C+SN+L still performs the best.  In some cases, the non-rigid transform learned by RCSLS may be inappropriate, or not as effective as in the XNLI task.  Overall, these extensive additional experiments with the additional I-C+L baseline further confirm the helpfulness of the Spectral Normalization.
>
>
> 3.  We attempted to quantify the significance of the improvement gained by Spectral Normalization.  While the numerical improvement of including this approach in addition to centering and length normalization is often small, it is quite consistent.  Measuring a statistical significance on any one task is difficult also since the techniques and experiments are deterministic.  However, we did show significance in the consistency of the improvement using a Binomial test.  This is shown in Appendix E.  For instance, in Table 15, using PROC-B alignment, I-C+SN+L normalization improves over I-C+L normalization on 25 of 28 languages.  Under the Binomial test, this has a p-value of $0.00001$, and is significant at reasonable significance levels (e.g., 0.01).

---

### Decision · Program_Chairs · 2022-01-20

**Decision:**

Accept (Poster)

**Comment:**

The authors propose a normalization method for cross-lingual text representations. The goal is to normalize the monolingual embeddings based on spectral normalization. The study shows that produced text representations keep their meaning and improve performance on downstream tasks.

There is a disagreement among the reviewers. The main concern is whether the main contribution is an empirical study or a novel idea.  I think the authors well-addressed the concerns of most reviewers. The idea and empirical study are enough for publication for ICLR-2022.